# An Enhanced Hunger Games Search Optimization with Application to Constrained Engineering Optimization Problems

**DOI:** 10.3390/biomimetics8050441

**Published:** 2023-09-20

**Authors:** Yaoyao Lin, Ali Asghar Heidari, Shuihua Wang, Huiling Chen, Yudong Zhang

**Affiliations:** 1Department of Computer Science and Artificial Intelligence, Wenzhou University, Wenzhou 325035, China; rainie1209@163.com (Y.L.); as_heidari@ut.ac.ir (A.A.H.); 2School of Computing and Mathematical Sciences, University of Leicester, Leicester LE1 7RH, UK; shuihuawang@ieee.org

**Keywords:** Hunger Games Search, swarm intelligence, logarithmic spiral, Rosenbrock Method, benchmark, engineering optimization problems

## Abstract

The Hunger Games Search (HGS) is an innovative optimizer that operates without relying on gradients and utilizes a population-based approach. It draws inspiration from the collaborative foraging activities observed in social animals in their natural habitats. However, despite its notable strengths, HGS is subject to limitations, including inadequate diversity, premature convergence, and susceptibility to local optima. To overcome these challenges, this study introduces two adjusted strategies to enhance the original HGS algorithm. The first adaptive strategy combines the Logarithmic Spiral (LS) technique with Opposition-based Learning (OBL), resulting in the LS-OBL approach. This strategy plays a pivotal role in reducing the search space and maintaining population diversity within HGS, effectively augmenting the algorithm’s exploration capabilities. The second adaptive strategy, the dynamic Rosenbrock Method (RM), contributes to HGS by adjusting the search direction and step size. This adjustment enables HGS to escape from suboptimal solutions and enhances its convergence accuracy. Combined, these two strategies form the improved algorithm proposed in this study, referred to as RLHGS. To assess the efficacy of the introduced strategies, specific experiments are designed to evaluate the impact of LS-OBL and RM on enhancing HGS performance. The experimental results unequivocally demonstrate that integrating these two strategies significantly enhances the capabilities of HGS. Furthermore, RLHGS is compared against eight state-of-the-art algorithms using 23 well-established benchmark functions and the CEC2020 test suite. The experimental results consistently indicate that RLHGS outperforms the other algorithms, securing the top rank in both test suites. This compelling evidence substantiates the superior functionality and performance of RLHGS compared to its counterparts. Moreover, RLHGS is applied to address four constrained real-world engineering optimization problems. The final results underscore the effectiveness of RLHGS in tackling such problems, further supporting its value as an efficient optimization method.

## 1. Introduction

Over the past few years, there have been remarkable advancements in optimization algorithms fueled by the rapid expansion of the engineering and artificial intelligence sectors [1,2]. These algorithms have broadened their capabilities to address intricate problem areas that conventional methods struggle with [3,4,5], including single-objective, multi-objective, and many-objective optimization problems [6,7]. They achieve this by utilizing evolutionary algorithms, swarm intelligence, and machine learning techniques. Integrating these algorithms with engineering and artificial intelligence has paved the way for hybrid approaches that harness the strengths of both disciplines. These advancements offer immense potential to tackle real-world challenges across diverse industries, enhance decision-making processes, optimize resource allocation, and improve overall system performance by delivering top-notch solutions for previously unsolvable problems [8,9]. 

Meta-heuristic Algorithms (MAs) have emerged as a crucial element in the field of artificial intelligence (AI), attracting substantial scholarly interest in the last two decades. This attention is primarily driven by their remarkable efficacy in addressing diverse practical problems, such as those encountered in economics management [10], power load forecasting [11], and data reduction problems [12,13,14,15], etc. The real-world optimization problems mentioned above generally share two common characteristics: high nonlinearity and inter-relation of decision variables during the solving process [16,17]. However, these characteristics inevitably give rise to large problem spaces, which make the optimization process prone to failure [18,19]. Therefore, it is crucial to consider factors such as time, risk, efficiency, and quality during the optimization process [20]. The emergence of Meta-heuristic Algorithms (MAs) has provided valuable inspiration to researchers in related fields. MAs offer reliability, robustness, and effective approaches for overcoming local optima by mimicking various phenomena to seek optimal solutions [21]. However, those metaphors do not change mathematics, and each optimizer’s core model will change its performance. The majority of Meta-heuristic Algorithms (MAs) derive their inspiration from natural phenomena and can be comprehensively classified into four distinct categories: evolutionary-based algorithms, swarm intelligence-based algorithms, human behavior-based algorithms, and physics-based algorithms. Evolutionary algorithms are meticulously devised based on the observable principles of evolution in nature. Swarm Intelligence (SI) algorithms draw inspiration from evolutionary theory and collective behavior, centering on the intricate behaviors displayed by systems comprising uncomplicated agents. Human behavior-based algorithms encompass diverse facets of human behavior, encompassing teaching, social interactions, learning, emotions, and management. Physics-based algorithms find inspiration in the laws of physics and mathematics. Table 1 provides some examples of related algorithm types.

Numerical optimization refers to finding the maximum or minimum objective value of a given problem within a defined search space. Population-based and derivative-free Swarm Intelligence (SI) optimization algorithms are widely regarded as effective solvers for complex numerical problems [37]. These algorithms utilize iterative methods to continually update individuals within a population, enhancing their adaptability to the environment and yielding acceptable optimal solutions within a reasonable timeframe. The study of SI optimization algorithms has received considerable attention in recent decades. Moreover, due to their superior performance, SI optimization algorithms also have been used to deal with multi-objective problems [38,39,40], constrained optimization problems [41,42,43,44], image segmentation [45,46,47,48], medical disease diagnosis [49,50,51,52,53], parameter estimation of solar photovoltaic models [54,55,56,57,58], intelligent traffic management [59], etc. Although not all the solutions obtained by these SI algorithms are optimal, what can be guaranteed is that high-quality solutions can be acquired in a reasonable time. 

Hunger Games Search (HGS) [60] is a novel algorithm proposed in 2021, designed based on natural animals’ foraging behavior. Once raised, HGS has received extensive attention from scholars. AbuShanab et al. [61] used the HGS optimizer to optimize a random vector functional link (RVFL) model, which successfully finds the optimal internal parameters of RVFL that boost the model accuracy. Nguyen et al. [62] combined HGS with Artificial Neural Network (ANN) named HGS-ANN for predicting ground vibration intensity problems. The experiment’s result verifies that HGS-ANN performs better than other same-type models. Like other SI algorithms, exploration and exploitation are two fundamental phases in HGS. The major work of the exploration phase is to search for a solution location, and the evaluation of promising solutions is completed within the exploitation stage. Through comparative research, it can be found that there are certain deficiencies in the performance of HGS in these two stages, which leads to premature and makes HGS prone to getting stuck at local optimal factors. To overcome the mentioned drawbacks, several authors enhanced HGS with efficient strategies. Xu et al. [63] improved HGS by incorporating the quantum rotation gate strategy and the Nelder-Mead simplex method, which explored the neighborhood of the optimal solution in the decision space more practically. For HGS is easily trapped in a local location and steadily converges speed while solving intricate problems. Ma et al. [64] introduced chaotic mappings, greedy selection, and vertical crossover strategy into the standard HGS. The introduced strategies were conducive to accelerating convergence velocity and enhancing search capability. A. Fathy et al. [65] introduced a non-homogeneous mutation operator to HGS, which proved effective in identifying the optimal settings for a Fractional-Order Proportional Integral Derivative (FOPID) based Load Frequency Controller (LFC). Emam et al. [66] modified the base HGS with a Local Escaping procedure with Brownian motion, to mitigate performance shortcomings. A. M. Nassef et al. [67] proposed a variant of HGS with a binary tau-based crossover plan. Zhang et al. [68] revealed improved HGS (IHGS) with cube mapping and refracted opposition-based learning policies. Besides, to recognize most important genes and handle the high-dimensional genetic data, Z. Chen et al. [69] invented a novel wrapper gene selection with artificial bee bare-bone Hunger Games Search (ABHGS). This variant combines HGS with an idea based on artificial bee mo-tions and a gaussian bare-bone idea.

However, based on the No Free Lunch (NFL) [70] theory, no such algorithm can universally solve all types of problems. In light of this, a new variant of the HGS algorithm is proposed in this study after identifying the limitations of the original HGS. To enhance the capability of exploration and exploitation, two adapted strategies are incorporated: the adapted Logarithmic Spiral strategy [71] (LS-OBL) and the adapted Rosenbrock Method (RM) [71]. The LS-OBL strategy is key in reducing the search space and maintaining solution diversity. By incorporating LS-OBL into HGS, the algorithm becomes more efficient in exploring different regions of the problem space and increasing the variety of solutions. On the other hand, the adapted RM strategy aids in overcoming local optima. It assists HGS in bypassing suboptimal solutions and improves its ability to converge toward better solutions.

The main contributions of this paper can be summarized as follows:The introduced strategy enhance the exploration and exploitation process of ordinary HGS algorithms when solving optimization problems.To evaluate the efficacy of the proposed approach, RLHGS is compared with eight other state-of-the-art algorithms on 23 classical benchmark functions and 10 benchmark functions from CEC2020. And the comparative evaluation of these experiments demonstrates the superiority of RLHGS in terms of optimization performance.The proposed RLHGS algorithm addresses four constrained real-world problems, showcasing its practical applicability and effectiveness in tackling complex engineering challenges.The experiment results of RLHGS indicate excellent accuracy and reliable performance.

The other part of this paper is organized as follows: Section 2 describes the standard HGS algorithm and the embedded strategies used in this study. Section 3 elaborates on the structure of the proposed RLHGS algorithm and displays its flowchart. Section 4 clarifies the experiment setting. Section 5 conducts a qualitative analysis and three experiments to demonstrate the improvement achieved by the embedded strategies, compare the performance of RLHGS with eight competing algorithms, and showcase its ability to handle practical engineering applications. Section 6 provides the conclusion and discusses the prospect of future work.

## 2. Preliminaries

### 2.1. Description of Hunger Games Search

Hunger serves as one of the most immediate homeostatic motivations in the lives of animals, influencing their behavioral decisions and actions. This fundamental motivation can even surpass and impact other competing drive states, such as thirst, feelings of insecurity, or fear of predators [72]. According to the literature [73], a conclusion can be drawn that with the increase in hunger, animal food cravings also increase. It is observed that hunger increases food craving in animals. In situations where there are limited food sources, a logical game emerges among hungry animals, where participants strive to secure victory and gain access to food sources for better chances of survival [74]. Building upon these premises, Hunger Games Search (HGS) was proposed.

#### 2.1.1. Approach Food

In nature, animals often engage in cooperative foraging behaviors, although this is not always the case [75]. There are instances where individuals choose to act alone. Based on studies on animal predatory behavior, Equation (1) introduces three position-updating modes that simulate the behavior of animals when they are in close proximity to food sources.
(1)$$\begin{array}{c} \vec{X\left(t+1\right)}=\left\{\begin{array}{c} {Game}_{1}:\vec{X\left(t\right)}\cdot \left(1+randn\left(1\right)\right),r_{1}<l\\ {Game}_{2}:\vec{W_{1}}\cdot \vec{X_{b}}+\vec{R}\cdot \vec{W_{2}}\cdot \left|\vec{X_{b}}-\vec{X\left(t\right)}\right|,r_{1}>l,r_{2}>E\\ {Game}_{3}:\vec{W_{1}}\cdot \vec{X_{b}}-\vec{R}\cdot \vec{W_{2}}\cdot \left|\vec{X_{b}}-\vec{X\left(t\right)}\right|,r_{1}>l,r_{2}<E \end{array}\right.\; \end{array}$$

In the above formula, t means current iterations and $\vec{X\left(t\right)}$ represents the position of each individual; $\vec{X_{b}}$ indicates the position of the best one in the current iteration; $randn\left(1\right)$ is a value that can satisfy normal distribution; $\vec{W_{1}}$ and $\vec{W_{2}}$ are indicators of hunger weight; $r_{1}$ and $r_{2}$ are two random values within the range of [0, 1]; l is a significant control parameter of the HGS, which can influence its overall performance. E is a variation control for all positions, the mathematical formula of it is as follows:(2)$$\begin{array}{c} E=\mathrm{sech}\left(\left|F\left(i\right)-BF\right|\right) \end{array}$$
where $F\left(i\right)$ records individual’s fitness value; BF means the best fitness acquired from the current iteration. What’s more, the specific expression of hyperbolic function sech in this study is as follows:(3)$$\begin{array}{c} \mathrm{sech}\left(x\right)=\frac{2}{e^{x}+e^{-x}} \end{array}$$

The formulas of $\vec{R}$ and relative parameters are as follows:(4)$$\begin{array}{c} R=2\times shrink\times rand-shrink \end{array}$$
(5)$$\begin{array}{c} shrink=2\times \left(1-\frac{t}{T}\right) \end{array}$$
where rand represents a random value limited within [0, 1]; T is the maximum value of iterations.

#### 2.1.2. Hunger Role

The starvation characteristic of individuals is the core content of the HGS algorithm. This subsection plans to present the mathematical model of this characteristic.
(6)$$\begin{array}{c} \vec{W_{1}\left(i\right)}=\left\{\begin{array}{c} hungey\left(i\right)\cdot \frac{N}{Sum\_hungry}\times r_{4},r_{3}<l\\ 1,\;r_{3}>l \end{array}\right. \end{array}$$
(7)$$\begin{array}{c} \vec{W_{2}\left(i\right)}=(1-\exp(-|hungry(i)|-Sum\_hungry))\times r_{5}\times 2 \end{array}$$

Equations (6) and (7) show how the weight value is calculated. In the formulas, hungry indicates the hunger condition of individuals, and Sum_hungry stands for the sum value of hungry; N is the total amount of individuals; $r_{3}$, $r_{4}$ and $r_{5}$ are random numbers which are limited in [0, 1]. The detailed expression of $hungry\left(i\right)$ is as follows:(8)$$\begin{array}{c} hungry\left(i\right)=\left\{\begin{array}{c} 0,\;AllFitness\left(i\right)==BF\\ hungry\left(i\right)+H,AllFitness\left(i\right)\;!=BF \end{array}\right. \end{array}$$
where the AllFitness stores all individual’s fitness value generated in iterations and the $AllFitness\left(i\right)$ indicates the fitness of each independent individual in the current iteration. Notably, when the best fitness is found, the corresponding $hungry\left(i\right)$ value would be assigned to 0, if not, the new hungry value H would be supplied based on the actual hungry value. Equation (9) showed as follows denotes the consists of H:(9)$$\begin{array}{c} H=\left\{\begin{array}{c} LH\times \left(1+r\right),TH<LH\\ TH,TH\geq LH \end{array}\right. \end{array}$$
(10)$$\begin{array}{c} TH=\frac{F\left(i\right)-BF}{WF-BF}\times r_{6}\times 2\times \left(UB-LB\right) \end{array}$$
where the sensation of hunger [76] H in Equation (9) is restricted to a lower bound LH which represents the lower limits of hunger. The setting of LH in this study is equal to 100, consisting with the settings in the literature [69,77]. BF and WF in Equation (10) means the best fitness value and the worst fitness value acquired from the current iteration, respectively; $F\left(i\right)$ stands for the fitness of each individual; $F\left(i\right)-BF$ denotes the threshold of food consumption necessary for an individual to attain a state of complete satiation; WF−BF denotes the maximal foraging ability of an individual in the current iteration; $\frac{F\left(i\right)-BF}{WF-BF}$ stands for the hunger ratio; $r_{6}$ is a random value limited within [0, 1]; UB and LB are the meanings of upper limits and lower limits of the dimensions, respectively. What’s more, in Equation (10), Algorithm 1. displays the pseudo-code of the HGS algorithm.

**Algorithm 1:** Pseudo-code of HGS.Initialize the parameters *N, T, l, D, Sum_hungry*Initialize the population $X_{i}\;(i=1,\;2,\;\ldots \;,\;n)$**While** (t ≤ T） Calculate the initial fitness of all populations Update *BF, WF*, and *X_b_* Calculate *hungry* by using Equation (8) Calculate *W*_1_ and *W*_2_ by using Equations (6) and (7), respectively **For**
*i* = 1 to *N*    **If (*rand* < 0.3)**     Update the position of the current search agent by using Equation (1)    **Else**     Calculate *E* by using Equation (2)     Update *R* using Equation (4)      Update the position of the current search agent by Equation (1)   **End if** **End For** *t* = *t* + 1
**End While**
Return *BF* and *X_b_*

### 2.2. The Adapted Logarithmic Spiral Strategy

Logarithmic Spiral (LS) [78] strategy, an effective search method in the exploration phase, is inspired by the spiral phenomena existing in nature [79]. Observing the famous Whale Optimization Algorithm (WOA) and Moth Flame Optimization (MFO) algorithm, the application of LS strategy can be found, which both use Equation (11) to mimic logarithmic spiral trajectory in their algorithm logic.
(11)$$\begin{array}{c} \vec{X(t+1)}=\left|\vec{X_{b}}-\vec{X(t)}\right|\times e^{br_{7}}\times \cos\left(2\pi r_{7}\right)+\vec{X_{b}} \end{array}$$

In Equation (11), $\vec{X_{b}}$ stands for the best solution location obtained from the current iteration; $\vec{X_{t}}$ and $\vec{X_{t+1}}$ means the position vector of t-th and (t+1)-th iteration, respectively; b is a constant value used to define the logarithmic spiral shape and the value of b is set to 1; $r_{7}$ is a random value range from [−1, 1]. 

An adapted LS strategy was proposed in the literature [71] to achieve a wider and more plausible range of exploration. This novel method combines the original LS strategy with Opposition-based Learning (OBL) [80], named LS-OBL strategy. The idea of this modified strategy is to incorporate the LS spatial trajectory between iteration-based and opposite-based solutions to boost the algorithm’s optimal efficiency. 

The implementation of the LS-OBL strategy is mainly divided into three parts. Firstly, the OBL algorithm generates an opposite solution $\vec{X_{op}}$ based on the $\vec{X_{b}}$ that the original HGS algorithm got in the current iteration. The mathematical model of this part is as follows:(12)$$\begin{array}{c} \vec{X_{op}}=rand\times \left(UB+LB\right)-\vec{X_{b}} \end{array}$$
where $\vec{X_{op}}$ is the opposite position vector of $\vec{X_{b}}$.

Then, the dynamic logarithmic spiral space between $\vec{X_{b}}$ and $\vec{X_{op}}$ is formed in each iteration. The relative mathematical model is mentioned in Equation (13).
(13)$$\begin{array}{c} \vec{X(t+1)}=\left|\vec{X_{b}}-\vec{X_{op}}\right|\times e^{br_{7}}\times \cos\left(2\pi r_{7}\right)+\vec{X_{b}} \end{array}$$

Lastly, the search agent achieves random exploration throughout the whole logarithmic spiral space according to the parameter of s which is defined in Equation (14).
(14)$$\begin{array}{c} s=2\times rand(0,\;1)-1 \end{array}$$
where rand(0, 1) means a random number generated from (0, 1).

### 2.3. The Adapted Rosenbrock Method Strategy

Rosenbrock Method (RM) [81] strategy is a reliable local search method proposed by Rosenbrock. For the poor performance of the basic RM strategy on multimodal problems [82], an enhanced variant based on the RM strategy was proposed by Li et al. [71] in 2021, which makes targeted improvement in the basic RM’s initial step size $\sigma_{i}(i=1,2,\ldots ,n)$ and termination conditions. Different from the $\sigma_{i}$ in tradition RM as a constant value, the $\sigma_{i}$ in adapted RM strategy can realize the dynamic change with iterations. The detailed description of $\sigma_{i}$ is as follows:(15)$$\begin{array}{c} \delta_{i}=\sqrt[{2}]{\left(\frac{\sum_{k=1}^{n}\left(\vec{X_{ki}}-\bar{\vec{X_{i}}}\right)}{n}\right)}+\varepsilon_{1},\;i=1,\;2,\;\ldots d \end{array}$$
(16)$$\begin{array}{c} \bar{\vec{X_{i}}}=\frac{\sum_{k=1}^{n}\vec{X_{ki}}}{n} \end{array}$$
where n represents the total amount of candidate individuals and d represents the dimension of the population. $\vec{X_{ki}}$ stands for the vector of the k-th individual in the i-th dimension, and $\bar{\vec{X_{i}}}$ stands for average candidate individual in the i-th dimension which is explained in Equation (16). $\varepsilon_{1}$ is a constant equal to $1.0e^{-150}$ for preventing the algorithm’s initial value from being 0.

What’s more, the two-loop termination condition in the original RM strategy is changed, which increases the participation of $\varepsilon_{1}$, $\varepsilon_{2}$, $k_{1}$, and $k_{2}$. Specifically, $\varepsilon_{1}$ and $\varepsilon_{2}$ are two parameters controlling internal and external loops, respectively. The parameter setting of $\varepsilon_{1}$ is mentioned before and the $\varepsilon_{2}$ is set to $1.0e^{-4}$, which values are consistent with the settings in the literature [82]. While the function of $k_{1}$ and $k_{2}$ play the role of loop counter. Algorithm 2 presents the pseudo-code of the improved RM strategy.
**Algorithm 2:** Pseudo-code of the adapted RM strategy.**Input** $\;\mathrm{the}\;\mathrm{search}\;\mathrm{agent}’s\;\mathrm{position}\;X_{i}\;(i=1,\;2,\;\ldots \;,\;n)$, population position X, D.$\mathrm{Initialize}\;\mathrm{the}\;\mathrm{orthonormal}\;\mathrm{basis}\;d_{i}\;(i=1,\;2,\;\ldots \;,\;n)$, the value of step size adjustment α and β$,\;\mathrm{the}\;\mathrm{ending}\;\mathrm{instruction}\;\varepsilon_{1},\;\varepsilon_{2},\;N,\;\;k_{2}=0$.
$\mathrm{Initialize}\;\mathrm{the}\;\mathrm{step}\;\mathrm{size}\;\delta_{i}\;(i=1,\;2,\;\ldots \;,\;n)$$\;\mathrm{by}\;\mathrm{using}\;\mathrm{Equation}\;(14),\;\mathrm{set}\;X_{k}=X_{i}$.
**While**$((\delta_{min}\geq \varepsilon_{1})\;or\;(k_{2}<2N$))  $\mathrm{Set}\;X=\;X_{k}$$,\;k_{1}=0$, Z=X   $\mathrm{While}\;(k_{1}<N$)     For i=1 to N      $Y=X+d_{i}\delta_{i}$      $\mathrm{If}\;(f\left(Y\right)<f(X)$)       X=Y       $\delta_{i}=a\delta_{i}\;(\alpha >1)$    **Else**       $\delta_{i}=\beta \delta_{i}\;(-1<\beta <0)$     **End If**    **End for**    $\mathrm{If}\;(\left[\frac{abs\left(f\left(Z\right)-f\left(X\right)\right)}{abs\left(f\left(X\right)\right)+\varepsilon_{1}}\right]<\varepsilon_{2})$      $k_{2}=k_{2}+1$    **Else**      $k_{2}=0$    **End If**    $k_{1}=k_{1}+1\;$   **End While**   $\mathrm{If}\;(f\left(X\right)<f(X_{k})$)     $X_{k}=X$      Update the orthonormal basis $d_{i}$.   **End If****End While****Return** $X_{b}$

## 3. Description of Proposed RLHGS

### 3.1. Motivation for This Work

Conventional swarm intelligence algorithms often suffer from issues such as premature convergence, slow convergence, and easy trapping in local optima. The design of the combination or hybrid algorithms can mitigate these problems to a certain extent. Maintaining the diversity of the population for simulation purposes and extending the flexibility of the algorithm to speed up the convergence velocity, hybridizing specific optimization strategies can make a good balance between the two phases of exploration and exploitation. 

There is no denying that Hunger Games Search (HGS) is a good population-based optimizer. However, when dealing with challenging optimization problems, the classic HGS sometimes shows premature convergence and stagnation shortcomings. Therefore, finding approaches that enhance solution diversity and exploitation capabilities is crucial. This study incorporates two effective strategies into HGS: adaptations based on the Logarithmic Spiral (LS-OBL) and Rosenbrock Method (RM). On the one hand, LS-OBL is an exploration method that is based on the idea of Logarithmic Spiral and Opposition-based Learning. Furthermore, the idea behind this strategy is to generate a batch of new solutions through the OBL strategy and then construct a logarithmic spiral spatial trajectory between the current solution and the OBL-based solution. LS-OBL effectively alleviates the defects of the classic HGS in exploration by properly narrowing space and increasing the solution’s diversity. On the other hand, the adapted RM method is employed to optimize the exploitation process. By adjusting the search direction and step size, RM helps the search agent avoid getting trapped into local optima, ensuring stronger convergence towards global optimal results. 

### 3.2. Flowchart and Pseudo-Code of RLHGS

The flowchart and pseudo-code are shown in Figure 1 and Algorithm 3, respectively. Notably, the execution of the RM strategy is conditional. For the high computational time of RM, the execution of RM is limited by a parameter prob, which plays the role of balancing RM performance and time consumption. The description of this parameter is as follows:(17)$$\begin{array}{c} {prob}_{i}=\frac{P_{no}}{N}\times rand(0,\;1)\;i=1,2,\;\ldots n\; \end{array}$$
where N means the size of the population; $P_{no}$ means the population number that is weaker than the optimal solution in the current iteration, rand is a random number obtained from (0, 1). When the $prob_{i}$ is higher than 0.8, the RM strategy will be invoked to search for the best further, or the exploitation strategy of standard HGS is performed.
**Algorithm 3:** Pseudo-code of RLHGS.Initialize the parameters N, T, l, D, Sum_hungry$\mathrm{Initialize}\;\mathrm{the}\;\mathrm{population}\;X_{i}\;(i=1,\;2,\;\ldots \;,\;n)$**While**  (t ≤ T) Calculate the initial fitness of all populations Update BF, WF$,\;\mathrm{and}\;X_{b}$ Calculate hungry by using Equation (8) $\mathrm{Calculate}\;W_{1}$ $\mathrm{and}\;W_{2}$ by using Equations (6) and (7), respectively **For** i=1 to N   **If (**rand
**< 0.3)**     Update the position of the current search agent by using the adapted LS-OBL strategy   **Else**     Calculate E by using Equation (2)     Update R using Equation (4)     Update the position of the current search agent by Equation (1)       **If (**prob
**> 0.8)**     Update the position of the current search agent by using the adapted RM strategy   **End If**   **End If**  **End For**  t=t + 1**End While****Return** BF$\;\mathrm{and}\;X_{b}$

### 3.3. Computational Complexity Analysis

Computational complexity is a measure of the time and resources required by an algorithm to execute. In the original HGS, the computational complexity mainly depends on these aspects: population initialization, fitness value calculation, sorting, and population updating. In these processes, N, D, and T represent the scale of population, the scale of dimension, and the maximum number of iterations respectively. Specifically, in the initial stage, the computational complexity of population initialization is O(N), whereas the computational complexity of fitness value calculation is O(T×N). In the worst case, the computational complexity of sorting is O(T×N×logN). The population updating includes hunger updating, weight updating, and location updating, with computational complexity O(T×N), O(T×N×D), and O(T×N×D), respectively. Thus, the computational complexity of the original HGS is O(N×(1+T×N×(2+logN+2×D))). The adapted algorithm RLHGS is also compounded from the above aspects, but owing to the addition of LS-OBL and RM operators, they differ in the process of updating population positions. However, due to the random operations of RM strategy, it is hard to assess the exact computational complexity of RLHGS. Hence, evaluating the algorithm’s computational cost necessitates taking into account the running time of the code. This study evaluates the actual computational cost of RLHGS and other comparative algorithms by recording their average time cost on 23 classic benchmark test suites. The average running cost comparison is listed in Section 5.3.1, and their units is seconds.

## 4. Designs for Experiments

### 4.1. Details of Benchmark Functions 

To mitigate the impact of randomness in algorithms, it is necessary to employ appropriate and comprehensive test functions and case studies. This ensures that superior results are not merely coincidental but consistently achieved. Thus, a sufficient evaluation is conducted using 23 classic benchmarks [83] and CEC 2020 benchmark test suite [84]. These benchmarks serve as crucial tools for testing algorithm performance. Twenty-three classical benchmark test suites consist of unimodal, multimodal, and fixed-dimensional multi-modal functions. Specifically, F1–F13 represent high-dimensional problems, including unimodal functions (F1–F5), a step function with one minimum value (F6), a noisy quartic function (F7), and multimodal functions with multiple local optima (F8–F13). Besides, F14–F23 are low-dimensional functions with only a few local minima, which enables the assessment of the algorithm’s effectiveness in searching for near-global optima. Detailed information about these benchmark functions can be found in Table 2. What’s more, to ascertain the RLHGS’s efficacy, it is also tested on the CEC2020 benchmark test suite, which includes one unimodal function, three multimodal functions, three hybrid functions, and three composition functions. Table 3 provides further details about the CEC2020 benchmark functions. Notably, both in Table 2 and Table 3, D means the dimensions of functions, R means the domain of functions, and $f_{min}$ means the optimum solution of the functions. Figure 2 presents 3-D map of some 23 classic benchmarks functions.

### 4.2. Configuration of Experiment Environment

In this study, two kinds of experiments on benchmark test suites are conducted, one aims to demonstrate the effectiveness of the added strategies and the other is focused on showcasing the superiority of RLHGS through a comparison with other powerful algorithms. To maintain fair comparisons, we have adhered to the suggested principles outlined in previous AI studies [85,86,87], which emphasize the importance of employing uniform conditions during the assessment of different methodologies [88,89]. Hence, the population size of these two experiments is set to 30, and the function evaluation for D = 30. The maximum iteration T is 300,000. To minimize the random error, all involved algorithms are run 30 times independently on all benchmark functions.

Besides, including the constrained real-world engineering experiment, all experiments are accomplished on a PC with Win11, a 64-bit operating system. The CPU is Intel (R) Core (TM) i5-9400, the main frequency is 2.90 GHz, the memory is 8.00 GB, and the software is MATLAB R2018b.

### 4.3. Statistical Analysis Methods

Evaluating the progress made by a new proposed algorithm compared to existing techniques is a specific challenge in experimental investigations. In recent years, researchers have recognized the importance of statistical analysis in assessing the performance of novel algorithms. In this study, the effectiveness of RLHGS is evaluated through several evaluation criteria. 

Firstly, the average value (Avg) and standard deviation (Std) of the optimal function value are used to evaluate the performance of algorithms. Among them, Avg is applied to evaluate the global search ability and the quality of the solution, while Std is devoted to evaluating the robustness of the algorithm. The ranking of each algorithm based on Avg is provided to reflect their performance on each function. In addition, the evaluation criteria of the Friedman test and Wilcoxon rank test also be used. In 1937, Milton Friedman first developed the concept of the Friedman test [90], which was later used to assess several different algorithms’ performance on different kinds of test functions. After its effectiveness is proven in various literature, the Friedman test has been regarded as an available method for model performance evaluation. Wilcoxon signed-rank test [91] was first proposed by a U.S. statistician named Frank Wilcoxon. As a hypothesis-testing method, the Wilcoxon rank test has been widely used to verify the algorithm’s statistical consistency after it was put forward. The principle of this method is to judge which algorithm is better by comparing the significant differences between two samples. Moreover, it can give a judgment if one algorithm is superior to another by calculating the p-value. Notably, the standard value of p-value is set to 0.05. The sign “+/−/=” in the table is utilized to indicate if the compared algorithm’s execution is better than, worse than, or comparable to that of RLHGS in terms of statistical manner. 

## 5. Result and Discussion

### 5.1. Qualitative Analysis

Exploration and exploitation are two crucial processes in SI algorithms. The exploration process primarily occurs in the early stages of algorithm execution, with the main aim of conducting a comprehensive search across the feasible domain space to identify potential regions and optimal solutions. During this phase, the algorithm would place significant emphasis on global search, which expands the search scope and diversifies search strategies to discover new areas that may contain better solutions. Exploration presents an opportunity to venture into uncharted territories and acts as a foundation for the subsequent exploitation process. The goal of the exploitation is to delve into known solution spaces and enhance the quality of candidate solutions through thorough searches. This stage focuses more on specific areas that are considered to have a higher probability of obtaining better solutions. By using more refined search strategies and utilizing prior knowledge, the exploitation phase can perform local optimizations around the current solution to obtain higher-quality solutions. Its major focus revolves around enhancing optimization performance to achieve swift convergence towards the optimal or approximate optimal solution.

To evaluate the exploration and exploitation processes of RLHGS, this subsection conducts a qualitative analysis. Figure 3 presents the qualitative results of RLHGS on several functions from the CEC2020 benchmark test suite, encompassing three types: unimodal function (F1), multimodal functions (F2 and F3), and composite function (F8). In Figure 3, column (a) presents the 3-D position distribution, showcasing the nature of four functions. Column (b) illustrates the 2-D spatial distribution of search history trajectories, providing insights into the position and dispersion of the population throughout the iteration process. The red dot in the image represents the global optimal value. Upon observing the graphs in this column, it becomes apparent that the population’s search trajectory almost revolves around the red dot. This observation suggests that the search range of RLHGS is both reasonable and effective. Column (c) showcases the motion trajectory of the first individual in the first dimension. It exhibits fluctuations during the initial stage of the search but ultimately converges towards the optimal value in later stages. This behavior can be attributed to the algorithm’s continuous pursuit of higher-quality solutions during the exploration phase, underscoring RLHGS’s adaptability and exceptional exploration capability. However, although the graphs in columns (b) and (c) indicate a trend for individuals of RLHGS to explore promising areas throughout the search space and ultimately utilize the best solution, the convergence curve has not been observed or proved. Column (d) records the convergence curves of RLHGS, revealing the trend of changes in the optimal fitness value and verifying the capability of RLHGS in obtaining a near-optimal solution throughout the whole iteration. 

However, when the processes of exploration and exploitation are not balanced, the optimization performance may not meet expectations. For instance, if the algorithm only possesses strong exploratory capabilities, it may yield high-quality solutions but at a slower convergence speed. On the other hand, if the algorithm leans towards exploitation, the convergence speed may improve, yet there is a higher risk of getting trapped in local optima. Hence, achieving a delicate balance between the exploration and exploitation stages becomes crucial for enhancing algorithm performance. 

To further examine the impact of LS-OBL and RM on the exploration and exploitation process. This study conducts a balance analysis and comprehensive discussions of these two processes of RLHGS and HGS. The relative results are shown in Figure 4. Notably, the % EPL and % EPT indicators in column (a) represent the proportions of algorithmic exploration and exploitation processes throughout the entire execution, which are calculated by Equations (18)–(20). In equations, Div refers to individual diversity, ${Div}_{max}$ indicates maximum individual diversity, ${Div}_{j}$ represents the j-th dimensional diversity of an individual, and n denotes the total number of individuals in the population. D represents the function’s dimension, $X_{ij}$ represents the j-th dimension of the i-th individual, and medium ($X_{j}$) signifies the median value of the j-th dimension across all individuals.
(18)$$\begin{array}{c} \%EPL=\frac{Div}{{Div}_{max}}\times 100\% \end{array}$$
(19)$$\begin{array}{c} \%EPT=\frac{\left|Div-{Div}_{max}\right|}{{Div}_{max}}\times 100\% \end{array}$$
(20)$$\begin{array}{c} {Div}_{j}=\frac{1}{n}\sum_{i=1}^{n}\left|median\left(x_{j}\right)-x_{ij}\right| \end{array}$$
(21)$$\begin{array}{c} Div=\frac{1}{dim}\sum_{j=1}^{dim}{Div}_{j} \end{array}$$

As shown in Figure 4, columns (a) and (b) illustrate the exploration and exploitation stage balance diagrams throughout the execution process, showcasing trend curves representing the exploration and exploitation stages. Except for the change curve of two processes, an incremental–decremental curve is added to reflect the algorithm’s level of exploration. If the global search outweighs local development during algorithm execution, the curve exhibits an upward trend. Conversely, if local development dominates, the curve demonstrates a downward trend. In detail, upon analyzing the graphs in columns (a) and (b), it becomes apparent that the exploration and exploitation process in RLHGS has shown significant improvement compared to HGS. On the F1 function, % EPL increased from 8.1273% to 16.7709%, indicating an improvement of 8.6517%. On the F2 function, % EPL increased from 8.6423% to 24.202%, representing a boost of 15.5597%. On the F3 function, % EPL rose from 3.1199% to 15.4554%, an increase of 12.3355%. Similarly, on the F8 function, % EPL surged from 6.125% to 22.7202%, marking an increase of 16.5952%. These numerical changes in % EPL and % EPT and the convergence curve separately generated by RLHGS and HGS in column (c) demonstrate that the inclusion of LS-OBL and RM algorithms has introduced a certain degree of balance between the exploration and exploitation stages.

### 5.2. Inspection of Improvement Effect

Even if the aforementioned results provide evidence and validation of RLHGS’s high performance, more specific information needs to be obtained to confidently confirm whether the added mechanism effectively promotes the performance of the HGS. In this experiment, there are four algorithms participating in the comparison. Table 4 lists an intuitive description of compared algorithms, where ‘1’ indicates that strategy is embedded, and ‘0’ indicates not to be adopted. All the algorithms are used to handle ten classical benchmark functions from CEC2020.

Table 5 presents the average value (Avg) and standard deviation (Std) of the optimal function value of RLHGS, RHGS, LHGS, and HGS. Upon observing ranking in Table 5, it is evident that RLHGS obtains the highest number of optimal values. Also, RLHGS exhibits a remarkable probability of approximately 80% in achieving the best performance across 10 test functions. Specifically, the outstanding performance of RLHGS is mainly reflected in the unimodal function (F1), multimodal functions (F2–F4), hybrid functions (F5–F7), and composition function (F8). Though slightly behind RHGS and HGS in F9 and F10, RLHGS still ranked first and obtained the lowest average value of the Friedman test shown in Table 6. Such a result not only indicates that RLHGS has good global search ability and higher quality solutions but also indicates that the robustness of the algorithm is better than compared algorithms. What’s more, it can be noticed that the original HGS is only in third place, which lags behind RLHGS in the ranking, directly verifying that embedded strategies have a good effect on promoting the optimization capability of HGS. 

A more intuitive comparison result can be observed from Figure 5. Looking at the convergence curves of F1, F2, F4, and F6, it can be concluded that neither the LS-OBL strategy nor the adapted RM strategy alone can effectively improve the HGS method sometimes, but when these two are simultaneously introduced, the improvement would be obvious. What’s more, the excellent performance of RLHGS can also be seen from the convergence curves of F3, F5, F7 and F8. The Wilcoxon signed-rank results in Table 7 also support the above conclusion.

In summary, the LS-OBL and adapted RM strategies work synergistically to overcome the limitations of the original algorithm, improving its overall performance in solving complex optimization problems.

### 5.3. Comparison with Eight Superior Algorithms

This subsection mainly introduces the experiment of RLHGS with eight state-of-the-art algorithms. Table 8 shows the parameter configuration of the algorithms involved in comparison and the brief introductions of these algorithms are as follows:CS [92]: Cuckoo search algorithm, a powerful algorithm that was presented by Gandomi et al. in 2013, the internal logic of the algorithm is based on the brood parasitism of cuckoo species.MFO [93]: Moth-flame optimization algorithm was a novel nature-inspired heuristic paradigm proposed by Mirjalili in 2015. The inspiration for designing this algorithm origins from the navigation method of moths in nature called transverse orientation.HHO [27]: Harris Hawks optimization algorithm was first proposed by Heidari et al. in 2019, simulating Harris hawks’ hunting behavior.SSA [94]: Salp Swarm Algorithm is a bio-inspired optimization algorithm that was developed by Mirjalili et al. in 2017. The idea is based on the swarming mechanism of salps.JADE [95]: An adaptive differential evolution algorithm, designed by Zhang et al. in 2009, implemented with a new mutation strategy IdquoDE/current-to-best duo with optional external archive and adaptively updating control parameters into normal differential evolution algorithm.ALCPSO [96]: An enhanced version of particle swarm optimization raised by Chen et al. in 2013, combined with an aging leader and challenger mechanism.SCGWO [97]: A variant of the grey wolf optimization algorithm innovated by Hu et al. in 2021, introduced the improved spread and chaotic local search strategies to the standard grey wolf optimization.RDWOA [98]: An improved meta-heuristic algorithm based on the original whale optimization algorithm developed in 2019, which is equipped with a random spare strategy and double adaptive weight.

#### 5.3.1. Benchmark Function Set I: 23 Classic Test Functions

To validate the feasibility of RLHGS, RLHGS with CS, MFO, HHO, SSA, JADE, ALCPSO, SCGWO, and RDWOA are arranged to handle 23 classical numerical optimization problems in this subsection. The comparison results are presented in Table 9. According to the Avg and Std, it can be found that RLHGS achieves the highest number of optimal values across various functions, including F1, F2, F9, F12, F13, F14, F16, F17, F19, F20, F21, F22 and F23. In contrast, other algorithms such as CS, MFO, SSA, JADE, and ALCPSO perform well only on fixed-dimensional multimodal functions, while HHO, SCGWO, and RDWOA show proficiency mainly in unimodal and multimodal functions. Only RLHGS consistently achieves ideal values across all function types, which indicates its versatility and robustness. Furthermore, Table 10 and Figure 6 provide the Friedman mean level and overall rank of all compared algorithms. Comparing these results, it is clear that RLHGS attains the highest rank, further solidifying its position as a powerful stochastic optimization algorithm. The Wilcoxon signed-rank results of RLHGS and other eight superior algorithms on 23 benchmark functions are shown in Table 11. In the table, the p-value which is less than 0.05, indicating a significant difference between RLHGS and the compared algorithm, with RLHGS performing better than the compared algorithm. Table 12 lists the average running time of RLHGS and other eight superior algorithms on 23 benchmark functions. Although the results are not the shortest time consuming, it can be seen that the algorithm has a relatively reasonable time cost on most functions.

Figure 7 portrays the convergence curves of this experiment. Look at Figure 7, when dealing with test functions F1 and F2, RLHGS obtains the optimum result with the fastest optimization speed. Moreover, when dealing with multimodal functions like F12 and F13, RLHGS is far more than other compared algorithms in searching for global or near-optimal solutions, which can be intuitively seen from the convergence curves that the algorithm does not immediately fall into local optima like other competing algorithms, also proving RLHGS can jump out of local optima. What’s more, for most test functions, the performance of the RLHGS is much better than other compared methods in the early search stage. Meanwhile, the final values obtained at the late search stage are faster or much closer to the optimal value. 

#### 5.3.2. Benchmark Function Set II: CEC2020 Test Functions

The evaluation configurations of this experiment are consistent with those in Section 5.3.1. Analyzing the comparison results revealed in Table 13, RLHGS outperforms all compared methods, which achieves five optimal values in ten test functions and no other algorithm exceeds it. In detail, it can be observed that RLHGS outperforms all of the competitors on multi-modal functions (F2–F3), hybrid function (F6), and composition function (F8), this result strongly reflects that RLHGS has advantage of exploration and local optima avoidance. Meanwhile, according to the statistical standard, it can be obtained that the function quantity of RLHGS superior to CS, MFO, ALCPSO, HHO, JADE, SCGWO, RDWOA, and SSA is 10, 10, 9, 7, 7, 7, 7, and 6, respectively, showing RLHGS is a competitive algorithm. Table 14 shows the result of the Friedman test result, where RLHGS secures the first position with a rank of 2.3, followed by JADE, SSA, HHO, RDWOA, and others. More intuitive ranking can be obtained from Figure 8. Table 15 shows the *p*-value results of the Wilcoxon signed-rank test. Upon observation, it is clear that RLHGS exhibits significant differences compared to other algorithms and exceeds them in terms of performance. Figure 9 shows four convergence curves of nine methods in this experiment, which are F2, F3, F4, and F6. The information that can be obtained from this figure is that RLHGS successfully exceeds other strong opponents and reaches a better solution.

Although the results and analyses in Section 5.3.1 and Section 5.3.2 verify that RLHGS has capability of determining the global optimal of the test functions to a certain degree, there still has some differences between actual problems and standard function problems. For example, the global optimal value for commonly used test functions is provided, whereas the global optimal value for actual problems remains unknown. What’s more, some equality and inequality constraints are also attached to practical problems. Therefore, in addition to the performance on the benchmark function, it is necessary to test the performance of the function in practical problems. In the next subsection, RLHGS is applied to solve four practical problems.

### 5.4. Four Real-World Constrained Benchmark Problems

In this subsection, the proposed RLHGS is used to settle four classical constrained benchmark problems, they are tension/compression spring design, welded beam design, pressure vessel design problem, and three-bar truss design. Due to their constraints being based on the different conditions, thus find a method that can effectively solve all these problems seems particularly significant. Researchers have recently proposed a mass of processing methods combining constraints with swarm intelligence algorithms. According to the different processing ways of these methods [99], the functions of the penalty are mainly divided into five categories: co-evolutionary, static, adaptive, dynamic, and death penalty functions. Considering the characteristic of the algorithm proposed, the method used in this study to handle four constrained benchmark problems is the death penalty function, which is the modest one in constructing an objective value of a mathematical model. 

#### 5.4.1. Tension/Compression String Problem

The way to solve the tension/compression string problem is to obtain the optimal parameters that can minimize the weight. This problem has three variables, which are wire diameter (d), mean coil diameter (D), and the number of active coils (N). Meanwhile, to solve this problem need to pay attention to four constraints functions $h_{1}\left(\vec{x}\right)$, $h_{2}\left(\vec{x}\right)$, $h_{3}\left(\vec{x}\right)$ and $h_{4}\left(\vec{x}\right)$. Its structure is shown in Figure 10. The mathematical description of this problem is shown as follows: 

Consider:$$\vec{x}=\left[x_{1},\;x_{2},\;x_{3}\right]=[d,\;D,\;N]$$

Objective function:$${f(\vec{x})}_{min}=x_{1}^{2}x_{2}x_{3}+2x_{1}^{2}x_{2}$$

Subject to
$$h_{1}\left(\vec{x}\right)=1-\frac{x_{2}^{3}x_{3}}{71,785x_{1}^{4}}\leq 0,$$
$$h_{2}\left(\vec{x}\right)=\frac{4x_{2}^{2}-x_{1}x_{2}}{12,566(x_{1}^{3}x_{2}-x_{1}^{4})}+\frac{1}{5108x_{1}^{2}}\leq 0,$$
$$h_{3}\left(\vec{x}\right)=1-\frac{140.45x_{1}}{x_{2}^{3}x_{3}}\leq 0,$$
$$h_{4}\left(\vec{x}\right)=\frac{x_{1}+x_{2}}{1.5}-1\leq 0$$

Variable ranges:$$0.05\leq x_{1}\leq 2.00,$$
$$0.25\leq x_{2}\leq 1.30,$$
$$2.00\leq x_{3}\leq 15.0$$

The comparison results of RLHGS with other eight advanced optimization algorithms in the tension/compression spring design problem are shown in Table 16. Observing the data from the table, it can be found that RLHGS obtains the lowest value 0.0126653 which is in bold, followed by INFO, IHS, PSO, and GSA close, which is 0.012666, 0.0126706, 0.0126747 and 0.0126763 respectively. The results verify that RLHGS has good capability in optimizing this engineering problem.

#### 5.4.2. Welded Beam Design Problem

The key to solving welded beam design problem is to acquire the optimal parameters that can minimize the cost of welded beams. In this problem, shear stress (τ), bending stress (θ), buckling load ($P_{c}$) and deflection (δ) are four constraints that need to be satisfied and thickness of welding seam (h), length of welding joint (l), width of beam (t), and thickness of bar (b) are four variables need to be considered. The shape of the welded beam design problem is shown in Figure 11. The following mathematical descriptions detailed describe this problem and its constraints: 

Consider:$$\vec{x}=\left[x_{1},\;x_{2},\;x_{3},\;x_{4}\right]=[h,\;l,\;t,\;b]$$

Minimize:$$f\left(\vec{x}\right)=1.10471x_{1}^{2}+0.04811x_{3}x_{4}(14.0+x_{4})$$

Subject to:$$g_{1}\left(\vec{x}\right)=\tau \left(\vec{x}\right)-\tau_{max}\leq 0$$
$$g_{2}\left(\vec{x}\right)=\sigma \left(\vec{x}\right)-\sigma_{max}\leq 0$$
$$g_{3}\left(\vec{x}\right)=\delta \left(\vec{x}\right)-\delta_{max}\leq 0$$
$$g_{4}\left(\vec{x}\right)=x_{1}-x_{4}\leq 0$$
$$g_{5}\left(\vec{x}\right)=P-P_{C}\left(\vec{x}\right)\leq 0$$
$$g_{6}\left(\vec{x}\right)=0.125-x_{1}\leq 0$$
$$g_{7}\left(\vec{x}\right)=1.10471x_{1}^{2}+0.04811x_{3}x_{4}(14.0+x_{2})-5.0\leq 0$$

Variable range: $$0.1\;\leq {\;x}_{1}\leq 2,\;0.1\;\leq \;x_{2}\leq 10,\;0.1\;\leq \;x_{3}\leq 10,\;0.1\;\leq \;x_{4}\leq 2$$
where:$$\tau \left(\vec{x}\right)=\sqrt{{\left(\tau^{\prime }\right)}^{2}+2\tau^{\prime }\tau^{\prime \prime }\frac{x_{2}}{2R}+{\left(\tau^{\prime \prime }\right)}^{2}},\tau^{\prime }=\frac{P}{\sqrt{2}x_{1}x_{2}}\;\tau^{\prime \prime }=\frac{MR}{J}\;M=P\left(L+\frac{x_{2}}{2}\right)$$
$$R=\sqrt{\frac{x_{2}^{2}}{4}+{\left(\frac{x_{1}+x_{3}}{2}\right)}^{2}}$$
$$J=2\left\{\sqrt{2}x_{1}x_{2}\left[\frac{x_{2}^{2}}{4}+{\left(\frac{x_{1}+x_{3}}{2}\right)}^{2}\right]\right\}$$
$$\sigma \left(\vec{x}\right)=\frac{6PL}{{x_{3}}^{2}x_{4}},\delta \left(\vec{x}\right)=\frac{6PL^{3}}{Ex_{3}^{2}x_{4}}$$
$$P_{C}\left(\vec{x}\right)=\frac{4.013E\sqrt{\frac{x_{3}^{2}x_{4}^{6}}{36}}}{L^{2}}\left(1-\frac{x_{3}}{2L}\sqrt{\frac{E}{4G}}\right)$$
$$P=6000\;lb,L=14\;in,\;\delta_{max}=0.25\;in,$$
$$E=30\times 1^{6}\;psi,G=12\times 10^{6\;}psi$$
$$\tau_{max}=13,600\;psi,\sigma_{max}=30,000\;psi$$

Table 17 shows the optimization results of welded beam design problem. This problem compares RLHGS with HGS, GSA, CDE, HS, GWO, BA, IHS, and RO. According to the data of optimum cost, the optimal value is shown in bold, which is obtained by RLHGS. Thus, it is easy to conclude that RLHGS performs best in solving the welded beam design problem. When the variables are set to 0.2015, 3.3345, 9.03662391, and 0.20572964, the optimum cost can reach 1.699986, lower than all compared algorithms. 

#### 5.4.3. Pressure Vessel Design Problem

Pressure vessel design problem is a conundrum in the engineering field. The way to solve this problem should focus on minimum the cost of welding, material, and forming of a vessel. The mathematical description of four variables and four constraints are shown in the following equations. Looking at these variables, $x_{1}$ to $x_{4}$ indicate the thickness of the shell ($T_{s}$), the thickness of the head ($T_{h}$), the internal radius (R), and the vessel length excluding head (L), respectively. Figure 12 displays the Components of this problem.

Consider:$$X=\left[x_{1},\;x_{2},\;x_{3},\;x_{4}\right]=[T_{s},\;T_{h},\;R,\;L]$$

Minimize:$$f(\vec{x})=0.6224x_{1}x_{3}x_{4}+1.7781x_{1}^{2}x_{3}+3.1661x_{1}^{2}x_{4}+19.84x_{1}^{2}x_{3}$$

Subject to:$$g_{1}\left(X\right)=-x_{1}+0.0193x_{3}\leq 0$$
$$g_{2}\left(X\right)=-x_{2}+0.00954x_{3}\leq 0$$
$$g_{3}\left(X\right)=-\pi x_{3}^{2}x_{4}-\frac{4}{3}\pi x_{3}^{3}+1,296,000\leq 0$$
$$g_{4}\left(X\right)=x_{4}-240\leq 0$$

Range of Variables:$$0\leq x_{1}\leq 99,$$
$$0\leq x_{2}\leq 99,$$
$$10\leq x_{3}\leq 200,$$
$$10\leq x_{4}\leq 200$$

In solving pressure vessel design problem, RLHGS is arranged to compare with ES, PSO, GA, G-QPSO, SMA, Branch-and-bound, HIS, GA3, and CPSO. The results in Table 18 show that when $T_{s}$, $T_{h}$, R, L are set as 0.8125, 0.4375, 42.0984456, 176.6365958 respectively, RLHGS gets the value 6059.714335, an optimal cost. This result demonstrates the proposed algorithm in this paper is superior to other algorithms for solving these kinds of mechanical engineering problems.

#### 5.4.4. Three-Bar Truss Design Problem

Three-bar truss design problem is a well-known constrained space problem, which is derived from civil engineering. Figure 13 presents its component. The method to solve this problem is to gain the minimum value of the weight of the bar structures. The stress constraints of each bar are the basis of the constraints in this problem. The problem is expressed mathematically in the following way:

Consider:$$f\left(x\right)=\left(2\sqrt{2}x_{1}+x_{2}\right)\times l$$

Subject to:$$g_{1}\left(x\right)=\frac{\sqrt{2}x_{1}+x_{2}}{\sqrt{2}x_{1}^{2}+2x_{1}x_{2}}P-\sigma \leq 0$$
$$g_{2}\left(x\right)=\frac{x_{2}}{\sqrt{2}x_{1}^{2}+2x_{1}x_{2}}P-\sigma \leq 0$$
$$g_{2}\left(x\right)=\frac{1}{\sqrt{2}x_{2}+x_{1}}P-\sigma \leq 0$$
where:$$0\leq x_{i}\leq 1\;i=1,\;2,\;3,$$
$$L\;=\;100\;\mathrm{cm},\;P\;=\;2\;\mathrm{kN}/{\mathrm{cm}}^{2},\;\sigma \;=\;2\;\mathrm{kN}/{\mathrm{cm}}^{2}$$

Table 19 lists the results of seven optimization algorithms in solving the three-bar design problem. In the table, optimum cost indicates the weight of the bar structures, and it is not hard to notice that RLHGS ranks first among all algorithms based on the minimum value of 263.89584338. BWOA and MVO ranked second and third with 263.8958435 and 263.8958499, respectively. Though the narrowing gap between values, it also supports the RLHSG can provide powerful assistance for dealing with three-bar design.

## 6. Conclusions and Future Work

HGS is a novel heuristic algorithm that has gained attention in recent years. Building upon the original literature, it is evident that HGS demonstrates remarkable optimization capabilities, surpassing numerous robust algorithms. However, the original HGS does have some limitations, including premature convergence, susceptibility to local optima, and slow convergence speed. These shortcomings indicate room for improvement within HGS. This study introduces the RLHGS algorithm, which incorporates an adapted LS-OBL mechanism and an adapted RM mechanism into the original HGS. These additions aim to enhance the algorithm’s exploration and exploitation abilities, respectively.

To assess the efficiency of these introduced mechanisms and the superiority of RLHGS over other powerful algorithms, several evaluations are conducted using the 23 classic benchmark functions and CEC2020 test suite, encompassing various function types. The first experiment analyzes the effectiveness of embedded strategies, yielding the conclusion that RLHGS performed exceptionally well when the LS-OBL strategy and adapted RM strategy worked in tandem. This finding validates the effectiveness of the added mechanisms in overcoming HGS’s drawbacks. In the second comparison experiment, RLHGS is compared alongside CS, MFO, HHO, SSA, JADE, ALCPSO, SCGWO, and RDWOA. According to the experiment results, the performance of RLHGS not only surpasses well-established classic algorithms like CS, SSA, JADE, RDWOA, and ALCPSO but also outperforms exceptional state-of-the-art algorithms such as MFO, HHO, and SCGWO. Furthermore, RLHGS is applied to optimize parameters in four engineering design problems. Comparative analysis with other algorithms reveals that the proposed method achieves superior results. Thus, RLHGS exhibits promise in tackling complex real-world optimization problems and could serve as a valuable auxiliary method for a broader range of global optimization problems. Overall, the integration of LS-OBL and RM into HGS, resulting in RLHGS, proves to be a valuable improvement, showcasing enhanced performance and robustness in various evaluation scenarios and real-world engineering optimization challenges. Additionally, this study only scratches one of RLHGS’s potential applications. In the future, RLHGS can find utility in numerous other fields beyond engineering optimization, such as image segmentation, machine learning model optimization, and others.

## Figures and Tables

**Figure 1 biomimetics-08-00441-f001:**
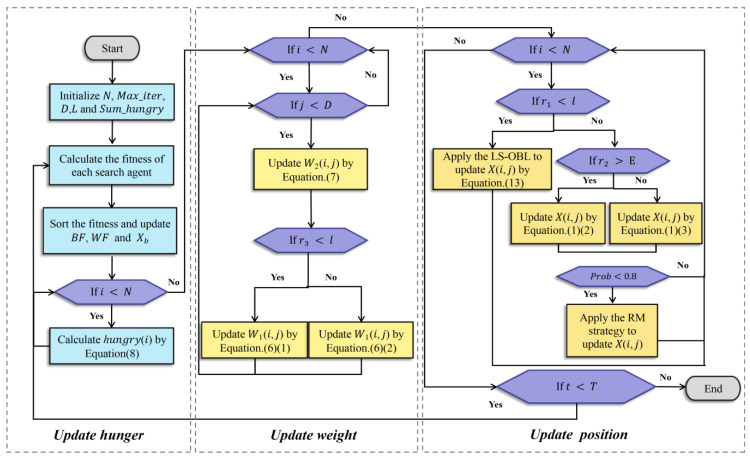
Flowchart of RLHGS.

**Figure 2 biomimetics-08-00441-f002:**
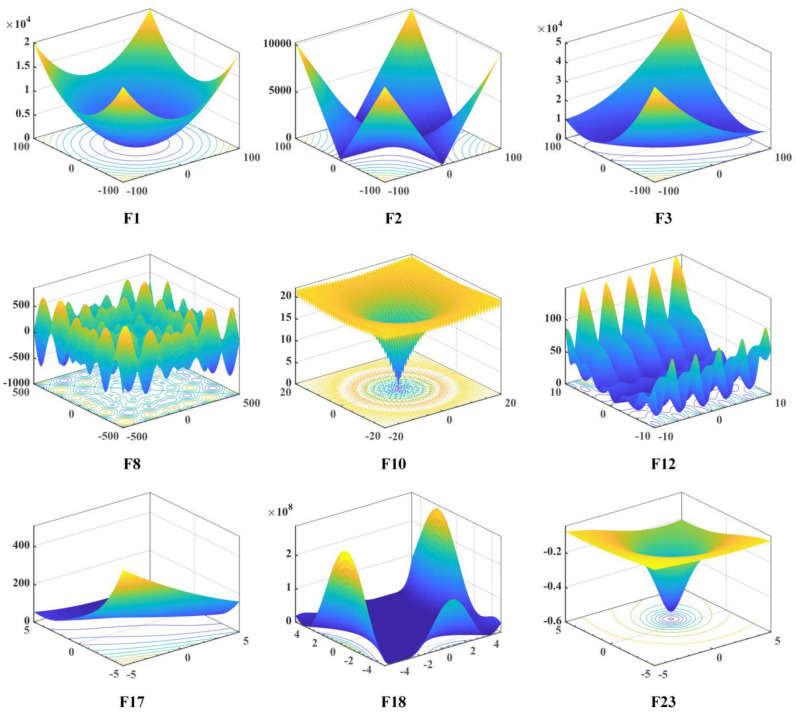
3-*D* map of some 23 classic benchmark functions. Different colors represent different solutions to the function, the mark of “F1”, “F2” etc. in the figure refer to the corresponding function.

**Figure 3 biomimetics-08-00441-f003:**
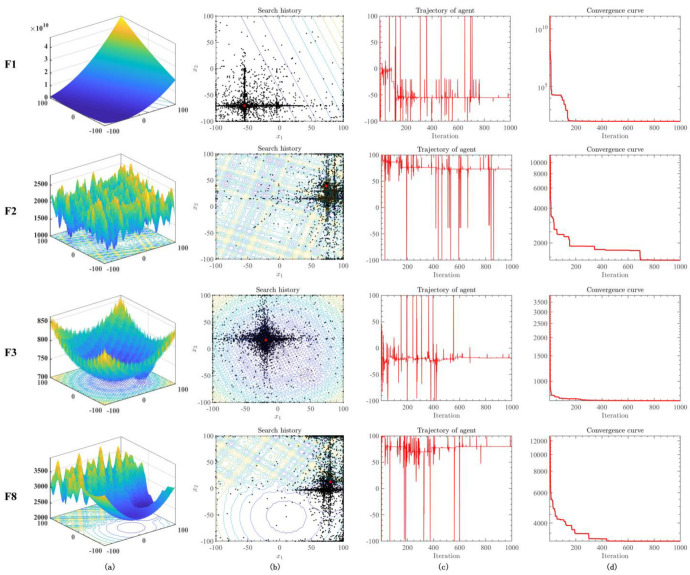
(**a**) 3-D distributions of test functions, (**b**) 2-D position distribution of RLHGS, (**c**) RLHGS trajectories in the first dimension, (**d**) convergence curve of RLHGS. The line is generated by the projection of a three-dimensional figure. Different color represents different solution of function.

**Figure 4 biomimetics-08-00441-f004:**
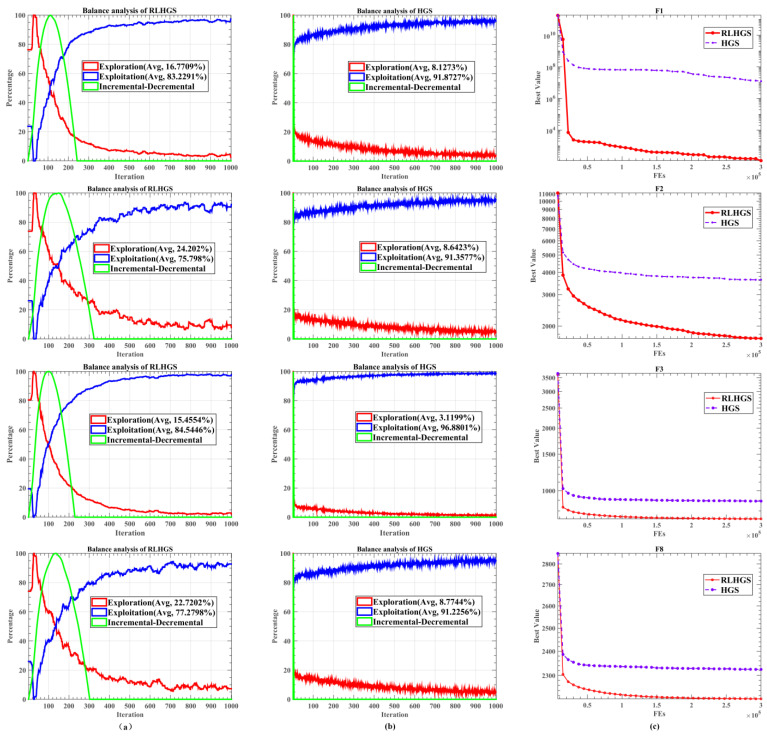
(**a**) balance analysis of the RLHGS, (**b**) balance analysis of the HGS, (**c**) convergence curves of RLHGS and HGS.

**Figure 5 biomimetics-08-00441-f005:**
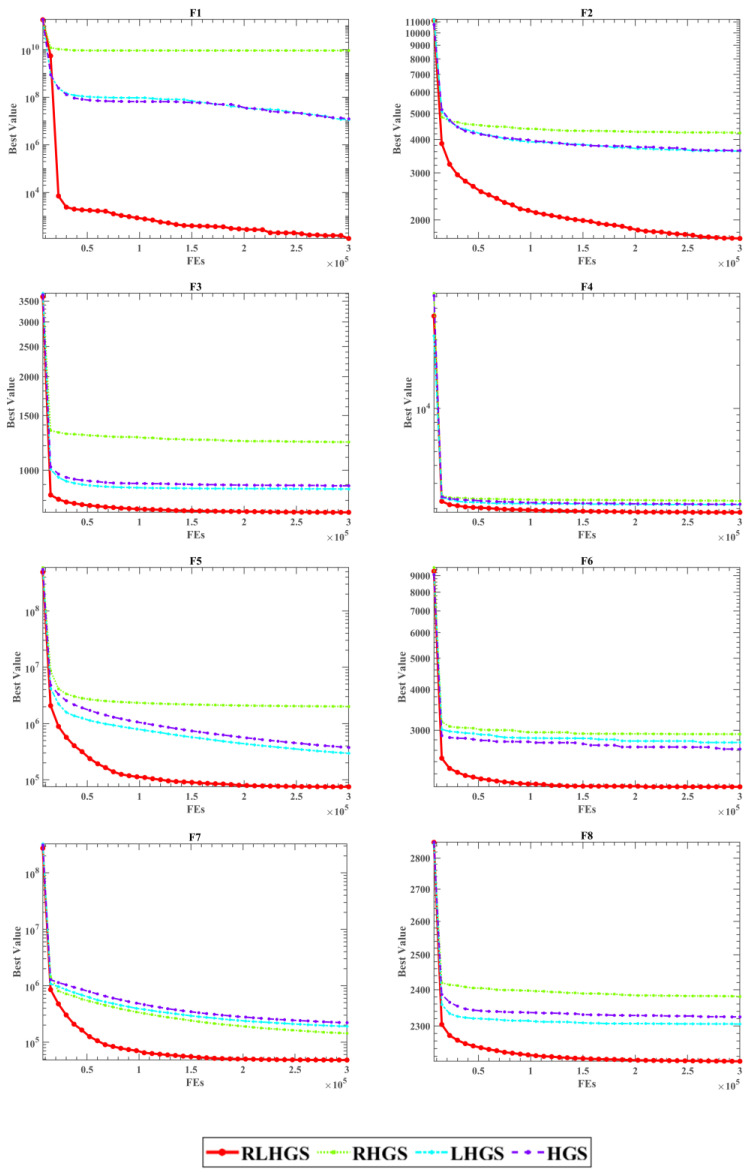
Convergence curves of RLHGS, RHGS, LHGS, and HGS on eight CEC2020 functions.

**Figure 6 biomimetics-08-00441-f006:**
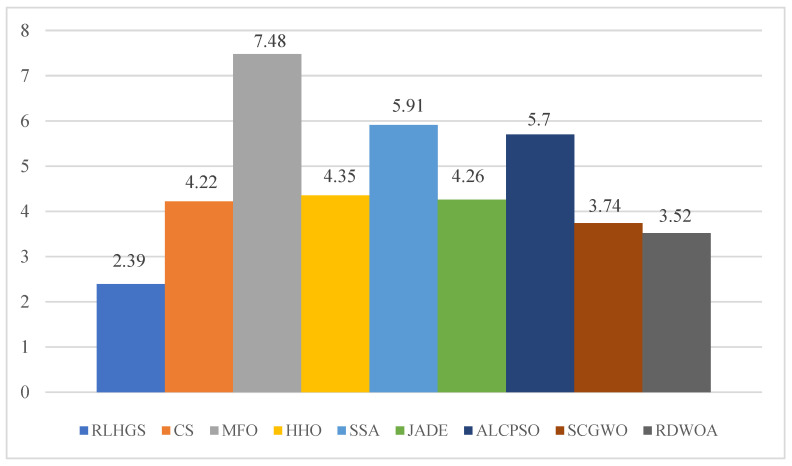
Friedman test results on 23 classic benchmark functions.

**Figure 7 biomimetics-08-00441-f007:**
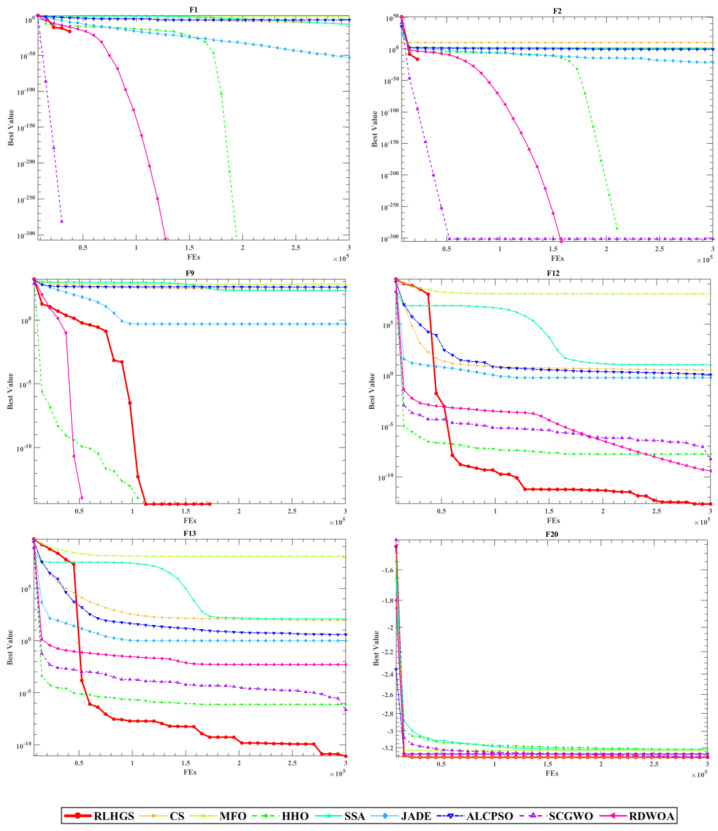
Convergence curves of compared algorithms on six classic benchmark functions.

**Figure 8 biomimetics-08-00441-f008:**
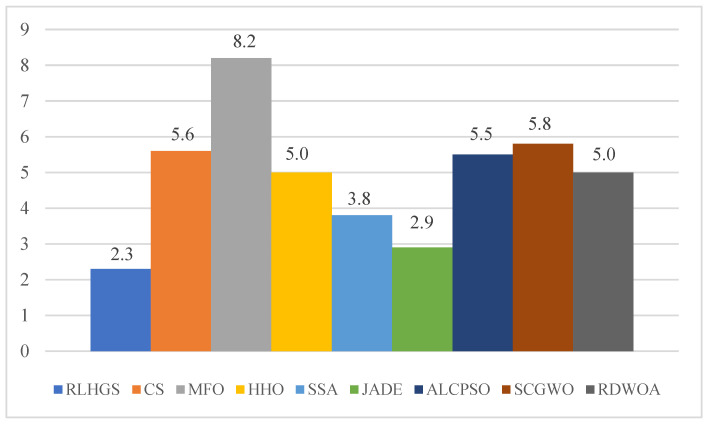
Friedman test results on CEC2020.

**Figure 9 biomimetics-08-00441-f009:**
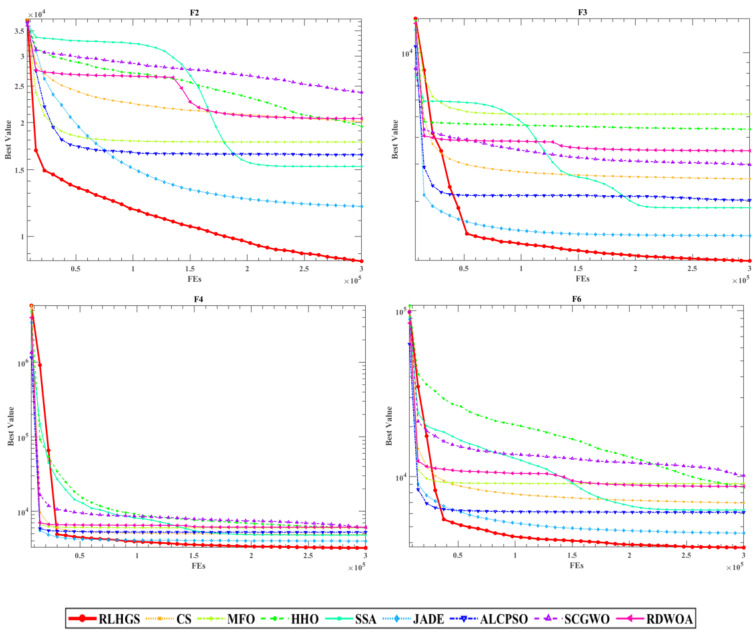
Convergence curves of compared algorithms on four CEC2020 functions.

**Figure 10 biomimetics-08-00441-f010:**
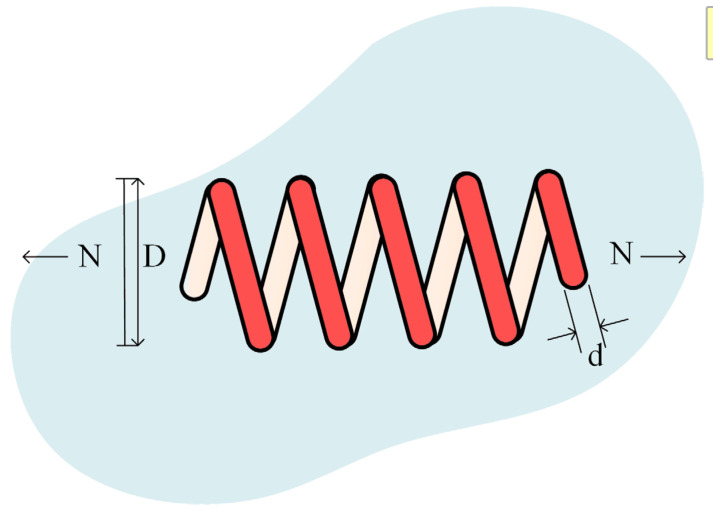
Structure of the tension/compression spring problem.

**Figure 11 biomimetics-08-00441-f011:**
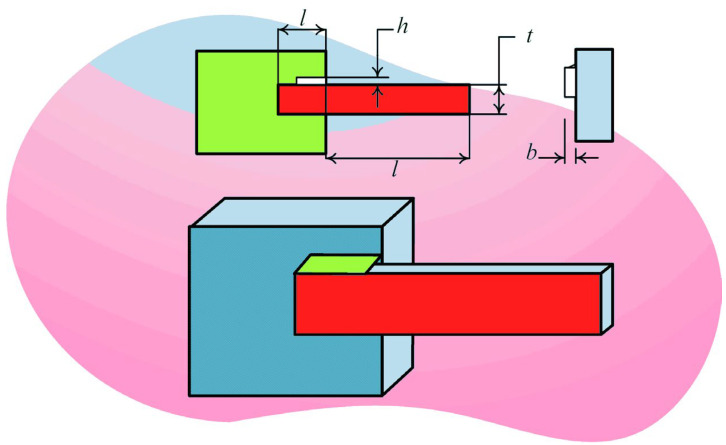
Shape of the welded beam design problem.

**Figure 12 biomimetics-08-00441-f012:**
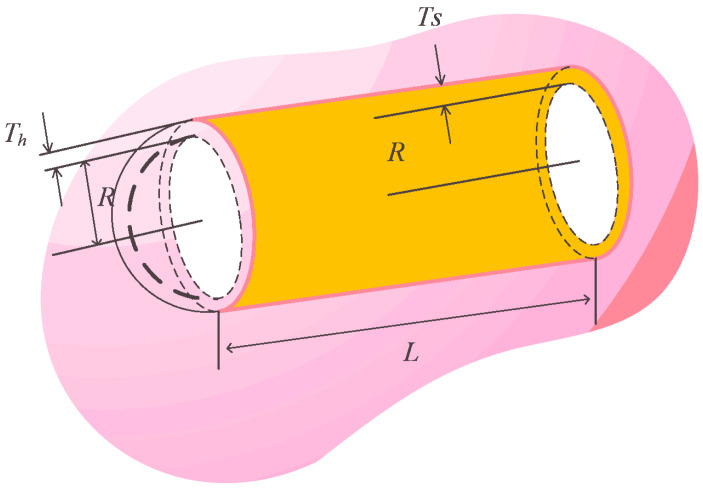
Components of the pressure vessel design problem.

**Figure 13 biomimetics-08-00441-f013:**
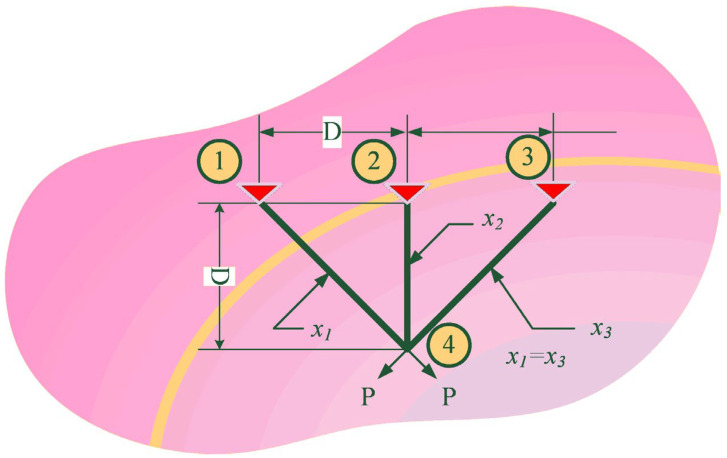
Components of the 3-bar truss design problem.

**Table 1 biomimetics-08-00441-t001:** Review of several classic optimization algorithms.

Type	MAs	Published	Brief Introduction
Evolutionary-based	Genetic Algorithm (GA) [22]	1975	It is derived from biological, genetic, and evolutionary mechanisms and an adaptive probabilistic optimization algorithm.
	Differential Evolution (DE) [23]	1995	It can be considered based on the theory of biological evolution, which imitates the process of cooperation and competition among individuals.
	Biogeography-Based Optimization (BBO) [24]	2008	It is based on the geographical distribution of biological organisms.
Swarm intelligence-based	Particle Swarm Optimization (PSO) [25]	1995	It is inspired by the collective behavior of social organisms, particularly the flocking and swarming behavior observed in birds, fish, and insects.
	Grey Wolf Optimization (GWO) [26]	2014	Its inspiration is from observing the leadership level and hunting behaviors within grey wolves in nature.
	Harris Hawk Optimization (HHO) [27]	2019	It draws upon the natural behavior of wolf pack hunting.
	Slime Mould Algorithm (SMA) [28]	2020	Its principle is based on the oscillation mode of slime moulds in nature.
Human behavior-based	Teaching-Learning-Based Optimization (TLBO) [29]	2011	It is inspired by the idea of how teachers guide students toward better learning outcomes.
	Social-Based Algorithm (SBA) [30]	2013	It is in the light of the evolutionary algorithm and socio-political process based Imperialist Competitive Algorithm (ICA) [31].
Physics-based	Simulated Annealing (SA) [32]	1983	It is proposed based on the principle of solid-state high-temperature annealing.
	Gravitational Search Algorithm (GSA) [33]	2009	It can trace back to the law of gravity and mass interactions.
	Multi-Verse Optimizer (MVO) [34]	2015	It is according to three cosmology concepts: white hole, black hole, and wormhole.
	RUNge Kutta Optimizer (RUN) [35]	2021	It combines elements of the classical Runge-Kutta numerical integration method with optimization techniques.
	weIghted meaN oF vectOrs (INFO) [36]	2022	It stems from the weight mean method, which is an enhanced optimizer in solving optimization problems.

**Table 2 biomimetics-08-00441-t002:** 23 classic benchmark functions.

Function	D	R	$f_{min}$
$F_{1}\left(x\right)=\sum_{i=1}^{n}x_{i}^{2}$	30	[−100, 100]	0
$F_{2}\left(x\right)=\sum_{i=1}^{n}\left|x_{i}\right|+\prod_{i=1}^{n}\left|x_{i}\right|$	30	[−10, 10]	0
$F_{3}\left(x\right)=\sum_{i=1}^{n}\;{\left(\sum_{j-1}^{i}\;x_{j}\right)}^{2}$	30	[−100, 100]	0
$F_{4}\left(x\right)={max}_{i}\left\{\left|x_{i}\right|,1\leq i\leq n\right\}$	30	[−100, 100]	0
$sF_{5}\left(x\right)=\sum_{i=1}^{n-1}\left[100{\left(x_{i+1}-x_{i}^{2}\right)}^{2}+{\left(x_{i}-1\right)}^{2}\right]$	30	[−30, 30]	0
$F_{6}\left(x\right)=\sum_{i=1}^{n}{\left([x_{i}+0.5]\right)}^{2}$	30	[−100, 100]	0
$F_{7}\left(x\right)=\sum_{i=1}^{n}ix_{i}^{4}+random[0,1]$	30	[−1.28, 1.28]	0
$F_{8}\left(x\right)=\sum_{i=1}^{n}{-x}_{i}\sin\left(\sqrt{\left|x_{i}\right|}\right)$	30	[−500, 500]	−418.9829 × 30
$F_{9}\left(x\right)=\sum_{i=1}^{n}\left[x_{i}^{2}-10\cos\left(2\pi x_{i}\right)+10\right]$	30	[−5.12, 5.12]	0
$F_{10}\left(x\right)=-20\exp\left(-0.2\sqrt{\frac{1}{n}\sum_{i=1}^{n}x_{i}^{2}}\right)-\exp\left(\frac{1}{n}\sum_{i=1}^{n}\cos\left(2\pi x_{i}\right)\right)+20+e$	30	[−32, 32]	0
$F_{11}\left(x\right)=\frac{1}{4000}\sum_{i=1}^{n}x_{i}^{2}-\prod_{i=1}^{n}\cos\left(\frac{x_{i}}{\sqrt{i}}\right)+1$	30	[−600, 600]	0
$F_{12}\left(x\right)=\frac{\pi }{n}\left\{10\;\sin\left(\pi y_{1}\right)+\sum_{i=1}^{n-1}{\left(y_{i}-1\right)}^{2}\left[1+10\sin^{2}\left(\pi y_{i+1}\right)\right]\;+{\left(y_{n}-1\right)}^{2}\right\}+\sum_{i=1}^{n}u\left(x_{i},10,100,4\right)\;y_{i}=1+\frac{x_{i}+1}{4}\;u\left(x_{i},a,k,m\right)=\left\{\begin{array}{c} \begin{array}{cc} k{\left(x_{i}-a\right)}^{m}\; & \;x_{i}>a \end{array}\\ \begin{array}{cc} 0\; & -a<x_{i}<a \end{array}\\ k{\left({-x}_{i}-a\right)}^{m}\;{\;x}_{i}<-a \end{array}\right.$	30	[−50, 50]	0
$F_{13}\left(x\right)=0.1\left\{\sin^{2}\left(3\pi x_{1}\right)+\sum_{i=1}^{n}{\left(x_{i}-1\right)}^{2}\left[1+\sin^{2}\left(3\pi x_{i}+1\right)\right]\;+{\left(x_{n}-1\right)}^{2}\left[1+\sin^{2}\left(2\pi x_{n}\right)\right]\right\}+\sum_{i=1}^{n}u\left(x_{i},5100,4\right)$	30	[−50, 50]	0
$F_{14}\left(x\right)={\left(\frac{1}{500}+\sum_{j=1}^{25}\frac{1}{j+\sum_{i=1}^{2}{\left(x_{i}-a_{ij}\right)}^{6}}\right)}^{-1}$	2	[−65, 65]	1
$F_{15}\left(x\right)=\sum_{i=1}^{11}\;{\left[a_{i}-\frac{x_{1}(b_{i}^{2}-b_{i}x_{2})}{b_{i}^{2}+b_{i}x_{3}+x_{4}}\right]}^{2}$	4	[−5, 5]	0.00030
$F_{16}\left(x\right)=4x_{1}^{2}-2.1x_{1}^{2}+\frac{1}{3}x_{1}^{6}+x_{1}x_{2}-4x_{2}^{2}+4x_{2}^{4}$	2	[−5, 5]	−1.0316
$F_{17}\left(x\right)={\left(x_{2}-\frac{5.1}{4\pi^{2}}x_{1}^{2}+\frac{5}{\pi }x_{1}-6\right)}^{2}+10\left(1-\frac{1}{8\pi }\right)\cos x_{1}+10$	2	[−5, 5]	0.398
$F_{18}\left(x\right)=\left[1+{\left(x_{1}+x_{2}+1\right)}^{2}\left(19-14x_{1}+3x_{1}^{2}-14x_{2}+6x_{1}x_{2}+3x_{2}^{2}\right)\right]\;\times \left[30+{\left(2x_{1}-3x_{2}\right)}^{2}\times (18-32x_{1}+12x_{1}^{2}+48x_{2}\;-36x_{1}x_{2}+27x_{2}^{2})\right]$	2	[−2, 2]	3
$F_{19}\left(x\right)=-\sum_{i=1}^{4}c_{i}exp\left(-\sum_{j=1}^{3}a_{ij}{\left(x_{j}-p_{ij}]\right)}^{2}\right)$	3	[1, 3]	−3.86
$F_{20}\left(x\right)=-\sum_{i=1}^{4}c_{i}exp\left(-\sum_{j=1}^{6}a_{ij}{\left(x_{j}-p_{ij}]\right)}^{2}\right)$	6	[0, 1]	−3.32
$F_{21}\left(x\right)=-\sum_{i=1}^{5}{\left[\left(X-a_{i}\right){\left(X-a_{i}\right)}^{T}+c_{i}\right]}^{-1}$	4	[0, 10]	−10.1532
$F_{22}\left(x\right)=-\sum_{i=1}^{7}{\left[\left(X-a_{i}\right){\left(X-a_{i}\right)}^{T}+c_{i}\right]}^{-1}$	4	[0, 10]	−10.4028
$F_{23}\left(x\right)=-\sum_{i=1}^{10}{\left[\left(X-a_{i}\right){\left(X-a_{i}\right)}^{T}+c_{i}\right]}^{-1}$	4	[0, 10]	−10.5363

**Table 3 biomimetics-08-00441-t003:** CEC2020 benchmark functions.

No.	Function	$f_{min}$
$F_{1}$	Shifted and Rotated Bent Cigar Function	100
$F_{2}$	Shifted and Rotated Schwefel’s Function	1100
$F_{3}$	Shifted and Rotated Lunacek bi-Rastrigin Function	700
$F_{4}$	Expanded Rosenbrock’s plus Criewangk’s Function	1900
$F_{5}$	Hybrid Function 1 (N = 3)	1700
$F_{6}$	Hybrid Function 2 (N = 4)	1600
$F_{7}$	Hybrid Function 3 (N = 5)	2100
$F_{8}$	Composition Function 1 (N = 3)	2200
$F_{9}$	Composition Function 2 (N = 4)	2400
$F_{10}$	Composition Function 3 (N = 5)	2500

**Table 4 biomimetics-08-00441-t004:** Design of RLHGS, RHGS, LHGS and HGS.

Algorithm	LS-OBL Strategy	Adapted RM Strategy
RLHGS	1	1
RHGS	0	1
LHGS	1	0
HGS	0	0

**Table 5 biomimetics-08-00441-t005:** Experiment results of RLHGS, RHGS, LHGS, and HGS on CEC2020.

	F1			F2			F3		
	Avg	Std	Rank	Avg	Std	Rank	Avg	Std	Rank
RLHGS	1.1721 × 10^2^	3.8415 × 10^1^	1	1.7057 × 10^3^	2.1073 × 10^2^	1	7.3244 × 10^2^	1.0536 × 10^1^	1
RHGS	9.1127 × 10^9^	1.3465 × 10^10^	4	4.2311 × 10^3^	6.1760 × 10^2^	4	1.2333 × 10^3^	1.6875 × 10^2^	4
LHGS	1.1411 × 10^7^	2.5083 × 10^7^	2	3.6214 × 10^3^	5.0565 × 10^2^	2	8.7070 × 10^2^	5.5928 × 10^1^	2
HGS	1.2428 × 10^7^	3.4529 × 10^7^	3	3.6354 × 10^3^	4.8399 × 10^2^	3	8.9153 × 10^2^	4.6783 × 10^1^	3
	F4			F5			F6		
	Avg	Std	Rank	Avg	Std	Rank	Avg	Std	Rank
RLHGS	1.8836 × 10^3^	8.9986 × 10^1^	1	7.4880 × 10^4^	4.3342 × 10^4^	1	2.0053 × 10^3^	1.5192 × 10^2^	1
RHGS	2.2603 × 10^3^	1.5553 × 10^2^	4	2.0131 × 10^6^	7.3692 × 10^6^	4	2.9175 × 10^3^	3.4464 × 10^2^	4
LHGS	2.1305 × 10^3^	1.9996 × 10^2^	2	2.9555 × 10^5^	2.0583 × 10^5^	2	2.7490 × 10^3^	2.8624 × 10^2^	3
HGS	2.1426 × 10^3^	1.5782 × 10^2^	3	3.7466 × 10^5^	2.8119 × 10^5^	3	2.6231 × 10^3^	2.8102 × 10^2^	2
	F7			F8			F9		
	Avg	Std	Rank	Avg	Std	Rank	Avg	Std	Rank
RLHGS	4.8111 × 10^4^	3.1322 × 10^4^	1	2.2069 × 10^3^	2.2097 × 10^0^	1	3.1351 × 10^3^	3.5632 × 10^2^	3
RHGS	1.4287 × 10^5^	1.2640 × 10^5^	2	2.3809 × 10^3^	2.9696 × 10^1^	4	2.6000 × 10^3^	7.2642 × 10^−13^	1
LHGS	1.8992 × 10^5^	1.3637 × 10^5^	3	2.3053 × 10^3^	3.2974 × 10^1^	2	3.1590 × 10^3^	4.0319 × 10^2^	4
HGS	2.2062 × 10^5^	1.7583 × 10^5^	4	2.3246 × 10^3^	3.3913 × 10^1^	3	2.6000 × 10^3^	0.0000 × 10^0^	1
	F10								
	Avg	Std	Rank	+/−/=					
RLHGS	2.8647 × 10^3^	7.4926 × 10^1^	4	~					
RHGS	2.7000 × 10^3^	4.3058 × 10^−13^	1	8/2/0					
LHGS	2.7797 × 10^3^	1.3715 × 10^2^	3	8/1/1					
HGS	2.7000 × 10^3^	0.0000 × 10^0^	1	8/2/0					

**Table 6 biomimetics-08-00441-t006:** The result of the Friedman test.

	RLHGS	RHGS	LHGS	HGS
Average rank	1.5	3.2	2.5	2.6
Overall rank	1	4	2	3

**Table 7 biomimetics-08-00441-t007:** Wilcoxon signed-rank results of RLHGS, RHGS, LHGS, and HGS on CEC 2020.

	RHGS	LHGS	HGS
F1	1.7344 × 10^−6^	1.7344 × 10^−6^	1.7344 × 10^−6^
F2	1.7344 × 10^−6^	1.7344 × 10^−6^	1.7344 × 10^−6^
F3	1.7344 × 10^−6^	1.7344 × 10^−6^	1.7344 × 10^−6^
F4	2.3534 × 10^−6^	3.7243 × 10^−5^	5.2165 × 10^−6^
F5	1.6046 × 10^−4^	1.9729 × 10^−5^	2.3534 × 10^−6^
F6	1.9209 × 10^−6^	1.7344 × 10^−6^	3.5152 × 10^−6^
F7	3.3173 × 10^−4^	9.3157 × 10^−6^	7.6909 × 10^−6^
F8	1.7344 × 10^−6^	1.7344 × 10^−6^	1.7344 × 10^−6^
F9	4.6072 × 10^−5^	1.5140 × 10^−1^	5.9493 × 10^−5^
F10	1.2290 × 10^−5^	2.0297 × 10^−3^	1.2290 × 10^−5^

**Table 8 biomimetics-08-00441-t008:** The parameter setting of the algorithms involved in the comparison.

Algorithm	Parameter	Value
RLHGS	l	0.03
	LH	100
	α	50
	β	−0.5
HGS	l	0.03
	LH	100
CS	N_iter	0
	pa	0.25
MFO	b	1
	t	[−2, 1]
HHO	beta	1.5
SSA	$c_{2}$	[0, 1]
	$c_{3}$	[0, 1]
JADE	c	0.1
	p	0.05
	CRm	0.5
	Fm	0.5
ALCPSO	w	0.4
	c1	2
	c2	2
	lifespan	60
	T	2
	pro	1/D
SCGWO	a	[2, 0]
	q	2
RDWOA	$a_{1}$	[2, 0]
	$a_{2}$	[−2, −1]
	b	1
	s	0

**Table 9 biomimetics-08-00441-t009:** Experiment results of RLHGS and other eight superior algorithms on 23 classic benchmark functions.

	F1			F2			F3		
	Avg	Std	Rank	Avg	Std	Rank	Avg	Std	Rank
RLHGS	0.0000 × 10^0^	0.0000 × 10^0^	1	0.0000 × 10^0^	0.0000 × 10^0^	1	4.8140 × 10^−1^	2.6367 × 10^0^	4
CS	1.0136 × 10^−6^	7.1003 × 10^−7^	8	1.0000 × 10^10^	0.0000 × 10^0^	9	2.7749 × 10^3^	3.6176 × 10^2^	7
MFO	1.9684 × 10^4^	1.1885 × 10^4^	9	1.4177 × 10^2^	3.8959 × 10^1^	8	1.2467 × 10^5^	7.6037 × 10^4^	9
HHO	0.0000 × 10^0^	0.0000 × 10^0^	1	0.0000 × 10^0^	0.0000 × 10^0^	1	0.0000 × 10^0^	0.0000 × 10^0^	1
SSA	7.0715 × 10^−8^	6.9886 × 10^−9^	7	6.0408 × 10^0^	2.7973 × 10^0^	7	1.8429 × 10^3^	5.7033 × 10^2^	6
JADE	3.7971 × 10^−54^	2.0335 × 10^−53^	5	7.8126 × 10^−21^	4.2783 × 10^−20^	5	5.1594 × 10^−1^	1.7506 × 10^0^	7
ALCPSO	5.5275 × 10^−8^	3.0275 × 10^−7^	6	2.8912 × 10^−1^	3.9540 × 10^−1^	6	4.3973 × 10^4^	4.9839 × 10^4^	8
SCGWO	0.0000 × 10^0^	0.0000 × 10^0^	1	1.6093 × 10^−306^	0.0000 × 10^0^	4	0.0000 × 10^0^	0.0000 × 10^0^	1
RDWOA	0.0000 × 10^0^	0.0000 × 10^0^	1	0.0000 × 10^0^	0.0000 × 10^0^	1	0.0000 × 10^0^	0.0000 × 10^0^	1
	F4			F5			F6		
	Avg	Std	Rank	Avg	Std	Rank	Avg	Std	Rank
RLHGS	1.3092 × 10^0^	3.6868 × 10^0^	4	4.9487 × 10^1^	3.4040 × 10^1^	3	4.6288 × 10^−11^	1.3740 × 10^−10^	2
CS	2.1938 × 10^1^	2.6548 × 10^0^	5	2.8015 × 10^2^	8.1279 × 10^1^	8	9.4781 × 10^−7^	7.5517 × 10^−7^	4
MFO	9.3119 × 10^1^	2.7711 × 10^0^	9	3.2477 × 10^7^	4.4852 × 10^7^	9	2.2485 × 10^4^	1.4372 × 10^4^	9
HHO	0.0000 × 10^0^	0.0000 × 10^0^	1	2.9834 × 10^−4^	4.0764 × 10^−4^	2	2.4328 × 10^−6^	3.4989 × 10^−6^	5
SSA	2.3199 × 10^1^	3.5560 × 10^0^	6	1.5513 × 10^2^	1.0766 × 10^2^	6	6.9376 × 10^−8^	6.2853 × 10^−9^	3
JADE	3.0183 × 10^1^	2.5466 × 10^0^	7	6.2184 × 10^1^	4.8003 × 10^1^	4	4.1087 × 10^−32^	2.7731 × 10^−32^	1
ALCPSO	4.6584 × 10^1^	5.0435 × 10^0^	8	1.7682 × 10^2^	5.6304 × 10^1^	7	2.5553 × 10^−5^	1.3996 × 10^−4^	8
SCGWO	0.0000 × 10^0^	0.0000 × 10^0^	1	6.7320 × 10^−5^	2.1095 × 10^−4^	1	4.6217 × 10^−6^	6.9114 × 10^−6^	6
RDWOA	0.0000 × 10^0^	0.0000 × 10^0^	1	9.0644 × 10^1^	4.7346 × 10^−1^	5	2.1185 × 10^−5^	1.1602 × 10^−4^	7
	F7			F8			F9		
	Avg	Std	Rank	Avg	Std	Rank	Avg	Std	Rank
RLHGS	8.4687 × 10^−2^	1.1647 × 10^−1^	5	−3.7931 × 10^4^	7.4989 × 10^3^	5	0.0000 × 10^0^	0.0000 × 10^0^	1
CS	4.2060 × 10^−1^	9.4582 × 10^−2^	7	−2.6888 × 10^4^	8.2358 × 10^2^	7	2.0495 × 10^2^	3.0866 × 10^1^	6
MFO	1.6836 × 10^2^	1.2589 × 10^2^	9	−2.4513 × 10^4^	2.3311 × 10^3^	9	6.5040 × 10^2^	9.4359 × 10^1^	9
HHO	1.3552 × 10^−5^	1.2912 × 10^−5^	1	−4.1898 × 10^4^	1.5437 × 10^−2^	2	0.0000 × 10^0^	0.0000 × 10^0^	1
SSA	1.4539 × 10^−1^	3.3744 × 10^−2^	6	−2.4613 × 10^4^	1.5061 × 10^3^	8	2.1037 × 10^2^	4.3408 × 10^1^	7
JADE	7.7322 × 10^−2^	2.2043 × 10^−2^	4	−4.0706 × 10^4^	3.5996 × 10^2^	4	1.3266 × 10^−1^	3.4400 × 10^−1^	5
ALCPSO	9.5572 × 10^−1^	4.4139 × 10^−1^	8	−3.2131 × 10^4^	1.4817 × 10^3^	6	3.5699 × 10^2^	5.1277 × 10^1^	8
SCGWO	1.6531 × 10^−5^	1.7151 × 10^−5^	2	−4.1898 × 10^4^	7.3117 × 10^−6^	1	0.0000 × 10^0^	0.0000 × 10^0^	1
RDWOA	1.6720 × 10^−5^	1.9081 × 10^−5^	3	−4.1681 × 10^4^	1.1514 × 10^3^	3	0.0000 × 10^0^	0.0000 × 10^0^	1
	F10			F11			F12		
	Avg	Std	Rank	Avg	Std	Rank	Avg	Std	Rank
RLHGS	1.5987 × 10^−15^	3.8918 × 10^−15^	4	1.1433 × 10^2^	4.1889 × 10^2^	8	8.6139 × 10^−14^	2.7494 × 10^−13^	1
CS	3.6675 × 10^0^	6.8688 × 10^−1^	8	1.4035 × 10^−3^	3.7653 × 10^−3^	4	2.6560 × 10^0^	8.6266 × 10^−1^	7
MFO	1.9796 × 10^1^	3.0301 × 10^−1^	9	1.4780 × 10^2^	1.5074 × 10^2^	9	1.1987 × 10^8^	1.6068 × 10^8^	9
HHO	8.8818 × 10^−16^	0.0000 × 10^0^	1	0.0000 × 10^0^	0.0000 × 10^0^	1	1.4939 × 10^−8^	2.4035 × 10^−8^	4
SSA	3.5158 × 10^0^	8.7325 × 10^−1^	7	2.9551 × 10^−3^	5.9380 × 10^−3^	5	1.1052 × 10^1^	2.8571 × 10^0^	8
JADE	3.0915 × 10^0^	7.0554 × 10^−1^	6	6.6576 × 10^−2^	2.2311 × 10^−1^	6	4.9293 × 10^−1^	8.7992 × 10^−1^	5
ALCPSO	3.0853 × 10^0^	1.0339 × 10^0^	5	1.4067 × 10^−1^	1.9612 × 10^−1^	7	1.1087 × 10^0^	1.4219 × 10^0^	6
SCGWO	8.8818 × 10^−16^	0.0000 × 10^0^	1	0.0000 × 10^0^	0.0000 × 10^0^	1	3.5795 × 10^−9^	6.2373 × 10^−9^	3
RDWOA	8.8818 × 10^−16^	0.0000 × 10^0^	1	0.0000 × 10^0^	0.0000 × 10^0^	1	3.7469 × 10^−10^	1.1328 × 10^−10^	2
	F13			F14			F15		
	Avg	Std	Rank	Avg	Std	Rank	Avg	Std	Rank
RLHGS	3.7509 × 10^−11^	1.8833 × 10^−10^	1	9.9800 × 10^−1^	0.0000 × 10^0^	1	3.3801 × 10^−4^	1.6718 × 10^−4^	5
CS	8.1878 × 10^1^	1.7632 × 10^1^	7	9.9800 × 10^−1^	0.0000 × 10^0^	1	3.0749 × 10^−4^	1.5595 × 10^−19^	1
MFO	1.9189 × 10^8^	3.1803 × 10^8^	9	1.7906 × 10^0^	1.2289 × 10^0^	9	1.1968 × 10^−3^	1.4423 × 10^−3^	9
HHO	1.3671 × 10^−6^	1.7078 × 10^−6^	3	9.9800 × 10^−1^	2.5569 × 10^−12^	8	3.1053 × 10^−4^	2.9635 × 10^−6^	4
SSA	1.2276 × 10^2^	2.8909 × 10^1^	8	9.9800 × 10^−1^	1.8895 × 10^−16^	1	7.0929 × 10^−4^	4.3532 × 10^−4^	7
JADE	1.1451 × 10^0^	1.8571 × 10^0^	5	9.9800 × 10^−1^	0.0000 × 10^0^	1	1.0676 × 10^−3^	3.6550 × 10^−3^	8
ALCPSO	3.6431 × 10^0^	6.1741 × 10^0^	6	9.9800 × 10^−1^	1.0100 × 10^−16^	1	3.6853 × 10^−4^	2.3232 × 10^−4^	6
SCGWO	3.0464 × 10^−7^	6.2419 × 10^−7^	2	9.9800 × 10^−1^	1.3287 × 10^−13^	6	3.1019 × 10^−4^	2.6873 × 10^−6^	3
RDWOA	8.2257 × 10^−3^	1.1120 × 10^−2^	4	9.9800 × 10^−1^	6.2046 × 10^−12^	7	3.0749 × 10^−4^	4.6780 × 10^−16^	2
	F16			F17			F18		
	Avg	Std	Rank	Avg	Std	Rank	Avg	Std	Rank
RLHGS	−1.0316 × 10^0^	6.7752 × 10^−16^	1	3.9789 × 10^−1^	0.0000 × 10^0^	1	3.0000 × 10^0^	2.0099 × 10^−15^	2
CS	−1.0316 × 10^0^	6.7752 × 10^−16^	1	3.9789 × 10^−1^	0.0000 × 10^0^	1	3.0000 × 10^0^	6.9974 × 10^−16^	1
MFO	−1.0316 × 10^0^	6.7752 × 10^−16^	1	3.9789 × 10^−1^	0.0000 × 10^0^	1	3.0000 × 10^0^	1.6941 × 10^−15^	4
HHO	−1.0316 × 10^0^	2.8301 × 10^−15^	7	3.9789 × 10^−1^	2.8584 × 10^−11^	7	3.0000 × 10^0^	2.1313 × 10^−12^	8
SSA	−1.0316 × 10^0^	5.4546 × 10^−16^	6	3.9789 × 10^−1^	6.1435 × 10^−16^	6	3.0000 × 10^0^	1.3515 × 10^−14^	7
JADE	−1.0316 × 10^0^	6.7752 × 10^−16^	1	3.9789 × 10^−1^	0.0000 × 10^0^	1	3.0000 × 10^0^	1.9039 × 10^−15^	2
ALCPSO	−1.0316 × 10^0^	5.9752 × 10^−16^	1	3.9789 × 10^−1^	0.0000 × 10^0^	1	3.0000 × 10^0^	1.8011 × 10^−15^	6
SCGWO	−1.0316 × 10^0^	1.9287 × 10^−6^	9	3.9796 × 10^−1^	8.3481 × 10^−5^	9	3.0000 × 10^0^	3.7102 × 10^−6^	9
RDWOA	−1.0316 × 10^0^	6.2844 × 10^−10^	8	3.9789 × 10^−1^	2.8285 × 10^−6^	8	3.0000 × 10^0^	2.0813 × 10^−15^	5
	F19			F20			F21		
	Avg	Std	Rank	Avg	Std	Rank	Avg	Std	Rank
RLHGS	−3.8628 × 10^0^	2.7101 × 10^−15^	1	−3.3220 × 10^0^	1.3424 × 10^−15^	1	−1.0153 × 10^1^	7.2269 × 10^−15^	1
CS	−3.8628 × 10^0^	2.7101 × 10^−15^	1	−3.3220 × 10^0^	1.2506 × 10^−15^	1	−1.0153 × 10^1^	7.2269 × 10^−15^	1
MFO	−3.8628 × 10^0^	2.7101 × 10^−15^	1	−3.2319 × 10^0^	7.0470 × 10^−2^	7	−7.7258 × 10^0^	3.1212 × 10^0^	8
HHO	−3.8628 × 10^0^	1.5442 × 10^−5^	7	−3.2245 × 10^0^	7.8815 × 10^−2^	8	−5.2251 × 10^0^	9.3075 × 10^−1^	9
SSA	−3.8628 × 10^0^	1.5668 × 10^−15^	6	−3.2190 × 10^0^	4.1107 × 10^−2^	9	−9.3111 × 10^0^	1.9151 × 10^0^	5
JADE	−3.8628 × 10^0^	2.7101 × 10^−15^	1	−3.2903 × 10^0^	5.3475 × 10^−2^	3	−8.8937 × 10^0^	2.3590 × 10^0^	6
ALCPSO	−3.8628 × 10^0^	2.5243 × 10^−15^	1	−3.2744 × 10^0^	5.9241 × 10^−2^	6	−8.7207 × 10^0^	2.4518 × 10^0^	7
SCGWO	−3.8606 × 10^0^	3.6749 × 10^−3^	9	−3.2902 × 10^0^	1.1989 × 10^−1^	4	−1.0153 × 10^1^	1.8808 × 10^−7^	4
RDWOA	−3.8625 × 10^0^	1.4390 × 10^−3^	8	−3.2840 × 10^0^	6.0187 × 10^−2^	8	−1.0153 × 10^1^	4.5944 × 10^−15^	1
	F22			F23					
	Avg	Std	Rank	Avg	Std	Rank	+/−/=		
RLHGS	−1.0403 × 10^1^	1.7140 × 10^−15^	1	−1.0536 × 10^1^	1.6820 × 10^−15^	1	~		
CS	−1.0403 × 10^1^	1.8067 × 10^−15^	1	−1.0536 × 10^1^	1.7455 × 10^−15^	1	12/3/8		
MFO	−8.5564 × 10^0^	3.1683 × 10^0^	8	−7.4807 × 10^0^	3.6232 × 10^0^	8	20/0/3		
HHO	−5.4420 × 10^0^	1.3483 × 10^0^	9	−5.4890 × 10^0^	1.3720 × 10^0^	9	12/6/5		
SSA	−1.0227 × 10^1^	9.6292 × 10^−1^	5	−1.0358 × 10^1^	9.7874 × 10^−1^	5	19/2/2		
JADE	−9.7180 × 10^0^	2.1204 × 10^0^	6	−9.7872 × 10^0^	2.2938 × 10^0^	7	10/2/11		
ALCPSO	−9.6985 × 10^0^	1.8230 × 10^0^	7	−1.0326 × 10^1^	9.9088 × 10^−1^	6	15/1/7		
SCGWO	−1.0403 × 10^1^	9.5393 × 10^−8^	4	−1.0536 × 10^1^	1.6328 × 10^−7^	4	12/6/5		
RDWOA	−1.0403 × 10^1^	7.6950 × 10^−6^	3	−1.0536 × 10^1^	1.3526 × 10^−5^	3	7/5/11		

**Table 10 biomimetics-08-00441-t010:** Friedman test results on 23 classic benchmark functions.

	RLHGS	CS	MFO	HHO	SSA	JADE	ALCPSO	SCGWO	RDWOA
Average rank	2.39	4.22	7.48	4.35	5.91	4.26	5.70	3.74	3.52
Overall rank	1	4	9	6	8	5	7	3	2

**Table 11 biomimetics-08-00441-t011:** Wilcoxon signed-rank results on 23 benchmark functions.

	CS	MFO	HHO	SSA	JADE	ALCPSO	SCGWO	RDWOA
F1	1.7344 × 10^−6^	1.7333 × 10^−6^	1.0000 × 10^0^	1.7333 × 10^−6^	1.7344 × 10^−6^	1.7344 × 10^−6^	1.0000 × 10^0^	1.0000 × 10^0^
F2	4.3205 × 10^−8^	1.7344 × 10^−6^	1.0000 × 10^0^	1.7344 × 10^−6^	1.7344 × 10^−6^	1.7344 × 10^−6^	1.2500 × 10^−1^	1.0000 × 10^0^
F3	1.7344 × 10^−6^	1.7344 × 10^−6^	3.9063 × 10^−3^	1.7344 × 10^−6^	3.1123 × 10^−5^	1.7344 × 10^−6^	3.9063 × 10^−3^	3.9063 × 10^−3^
F4	1.7344 × 10^−6^	1.7344 × 10^−6^	3.7896 × 10^−6^	1.7344 × 10^−6^	1.7344 × 10^−6^	1.7344 × 10^−6^	3.7896 × 10^−6^	3.7896 × 10^−6^
F5	1.7344 × 10^−6^	1.7344 × 10^−6^	1.9209 × 10^−6^	1.9209 × 10^−6^	3.0861 × 10^−1^	1.7344 × 10^−6^	1.7344 × 10^−6^	9.3157 × 10^−6^
F6	1.7344 × 10^−6^	1.7333 × 10^−6^	1.7344 × 10^−6^	1.7344 × 10^−6^	1.7333 × 10^−6^	3.7243 × 10^−5^	1.7344 × 10^−6^	1.7344 × 10^−6^
F7	2.3534 × 10^−6^	1.7344 × 10^−6^	1.7344 × 10^−6^	1.2453 × 10^−2^	4.5281 × 10^−1^	1.7344 × 10^−6^	1.7344 × 10^−6^	1.9209 × 10^−6^
F8	3.1123 × 10^−5^	1.9729 × 10^−5^	4.4919 × 10^−2^	3.1123 × 10^−5^	1.0201 × 10^−1^	1.0570 × 10^−4^	5.7064 × 10^−4^	6.0350 × 10^−3^
F9	1.7344 × 10^−6^	1.7344 × 10^−6^	1.0000 × 10^0^	1.7344 × 10^−6^	7.8125 × 10^−3^	1.7344 × 10^−6^	1.0000 × 10^0^	1.0000 × 10^0^
F10	1.7344 × 10^−6^	1.7344 × 10^−6^	1.0000 × 10^0^	1.7344 × 10^−6^	1.7344 × 10^−6^	1.7344 × 10^−6^	1.0000 × 10^0^	1.0000 × 10^0^
F11	4.9498 × 10^−2^	3.5876 × 10^−4^	6.2500 × 10^−2^	4.0702 × 10^−2^	3.6811 × 10^−2^	1.4793 × 10^−2^	6.2500 × 10^−2^	6.2500 × 10^−2^
F12	1.7344 × 10^−6^	1.7344 × 10^−6^	1.7344 × 10^−6^	1.7344 × 10^−6^	1.1499 × 10^−4^	1.7344 × 10^−6^	1.7344 × 10^−6^	1.7344 × 10^−6^
F13	1.7344 × 10^−6^	1.7344 × 10^−6^	1.7344 × 10^−6^	1.7344 × 10^−6^	2.0589 × 10^−1^	1.9209 × 10^−6^	1.9209 × 10^−6^	1.7344 × 10^−6^
F14	1.0000 × 10^0^	4.8828 × 10^−4^	1.7344 × 10^−6^	1.0000 × 10^0^	1.0000 × 10^0^	1.0000 × 10^0^	1.7213 × 10^−6^	1.5625 × 10^−2^
F15	8.4303 × 10^−6^	1.7257 × 10^−6^	3.1123 × 10^−5^	1.0246 × 10^−5^	2.2513 × 10^−2^	1.7372 × 10^−1^	3.1123 × 10^−5^	3.1123 × 10^−5^
F16	1.0000 × 10^0^	1.0000 × 10^0^	4.8828 × 10^−4^	2.2090 × 10^−5^	1.0000 × 10^0^	1.0000 × 10^0^	1.7344 × 10^−6^	1.0000 × 10^0^
F17	1.0000 × 10^0^	1.0000 × 10^0^	4.0100 × 10^−5^	1.2500 × 10^−1^	1.0000 × 10^0^	1.0000 × 10^0^	1.7344 × 10^−6^	4.8828 × 10^−4^
F18	5.7330 × 10^−7^	1.6244 × 10^−4^	1.6837 × 10^−6^	1.5871 × 10^−6^	3.4375 × 10^−1^	6.1035 × 10^−5^	1.7344 × 10^−6^	1.4307 × 10^−1^
F19	1.0000 × 10^0^	1.0000 × 10^0^	1.7344 × 10^−6^	1.2207 × 10^−4^	1.0000 × 10^0^	1.0000 × 10^0^	1.7344 × 10^−6^	1.2500 × 10^−1^
F20	1.0000 × 10^0^	4.3895 × 10^−5^	1.7344 × 10^−6^	1.7322 × 10^−6^	7.8125 × 10^−3^	4.8828 × 10^−4^	1.7344 × 10^−6^	9.7656 × 10^−4^
F21	1.0000 × 10^0^	4.8828 × 10^−4^	1.7344 × 10^−6^	1.7333 × 10^−6^	1.5625 × 10^−2^	7.8125 × 10^−3^	1.7344 × 10^−6^	5.0000 × 10^−1^
F22	1.0000 × 10^0^	7.8125 × 10^−3^	1.7344 × 10^−6^	1.7344 × 10^−6^	2.5000 × 10^−1^	6.2500 × 10^−2^	1.7344 × 10^−6^	1.2500 × 10^−1^
F23	1.0000 × 10^0^	2.4414 × 10^−4^	1.7344 × 10^−6^	1.7322 × 10^−6^	2.5000 × 10^−1^	2.5000 × 10^−1^	1.7344 × 10^−6^	1.2500 × 10^−1^

**Table 12 biomimetics-08-00441-t012:** Execution time of RLHGS and other eight superior algorithms on 23 benchmark functions.

	RLHGS	CS	MFO	HHO	SSA	JADE	ALCPSO	SCGWO	RDWOA
F1	8.4691 × 10^2^	2.0354 × 10^0^	1.4498 × 10^0^	1.4526 × 10^0^	1.2668 × 10^0^	1.6710 × 10^1^	8.3820 × 10^−1^	1.9997 × 10^0^	1.0154 × 10^0^
F2	3.0396 × 10^2^	2.0626 × 10^0^	1.5539 × 10^0^	1.2214 × 10^0^	1.1745 × 10^0^	1.6865 × 10^1^	7.9440 × 10^−1^	1.1329 × 10^1^	8.6280 × 10^−1^
F3	2.0991 × 10^3^	4.2184 × 10^0^	3.6022 × 10^0^	3.8757 × 10^0^	3.6416 × 10^0^	1.6792 × 10^1^	2.9570 × 10^0^	4.5626 × 10^0^	3.4932 × 10^0^
F4	7.0378 × 10^0^	1.9587 × 10^0^	1.4081 × 10^0^	1.1868 × 10^0^	1.1103 × 10^0^	1.6182 × 10^1^	7.3280 × 10^−1^	1.9553 × 10^0^	9.5240 × 10^−1^
F5	4.4681 × 10^0^	2.2221 × 10^0^	1.6676 × 10^0^	1.5862 × 10^0^	1.4441 × 10^0^	1.2135 × 10^1^	1.0096 × 10^0^	2.1779 × 10^0^	1.0915 × 10^0^
F6	9.6883 × 10^0^	1.9612 × 10^0^	1.3927 × 10^0^	1.2491 × 10^0^	1.1359 × 10^0^	1.3613 × 10^1^	7.6700 × 10^−1^	1.8489 × 10^0^	7.8840 × 10^−1^
F7	8.3248 × 10^0^	3.1648 × 10^0^	2.6076 × 10^0^	2.4620 × 10^0^	2.4082 × 10^0^	1.3690 × 10^1^	1.9893 × 10^0^	3.1472 × 10^0^	2.0583 × 10^0^
F8	1.5842 × 10^1^	2.4678 × 10^0^	1.6926 × 10^0^	1.6836 × 10^0^	1.4959 × 10^0^	1.2785 × 10^1^	1.0662 × 10^0^	2.2461 × 10^0^	1.1192 × 10^0^
F9	9.9402 × 10^0^	2.1947 × 10^0^	1.6374 × 10^0^	1.4256 × 10^0^	1.3468 × 10^0^	1.3356 × 10^1^	9.2490 × 10^−1^	1.9687 × 10^0^	8.8810 × 10^−1^
F10	1.9390 × 10^3^	2.1445 × 10^0^	1.5712 × 10^0^	1.4695 × 10^0^	1.3655 × 10^0^	1.3987 × 10^1^	1.0166 × 10^0^	1.9999 × 10^0^	8.9220 × 10^−1^
F11	2.0717 × 10^1^	2.2916 × 10^0^	1.9069 × 10^0^	1.6695 × 10^0^	1.6033 × 10^0^	1.4202 × 10^1^	1.2030 × 10^0^	2.2250 × 10^0^	1.1195 × 10^0^
F12	9.4747 × 10^1^	5.3738 × 10^0^	4.9777 × 10^0^	5.0041 × 10^0^	4.8790 × 10^0^	1.3993 × 10^1^	4.3605 × 10^0^	5.5637 × 10^0^	4.4780 × 10^0^
F13	9.8841 × 10^1^	5.3597 × 10^0^	4.8539 × 10^0^	4.9606 × 10^0^	4.9572 × 10^0^	1.4223 × 10^1^	4.2401 × 10^0^	5.5618 × 10^0^	4.4716 × 10^0^
F14	1.5509 × 10^1^	7.4750 × 10^0^	7.0009 × 10^0^	7.9889 × 10^0^	7.4952 × 10^0^	1.5530 × 10^1^	7.0055 × 10^0^	7.4837 × 10^0^	7.2453 × 10^0^
F15	1.0824 × 10^0^	1.3539 × 10^0^	7.2870 × 10^−1^	9.9650 × 10^−1^	7.6850 × 10^−1^	1.4078 × 10^1^	5.9570 × 10^−1^	7.4800 × 10^−1^	5.1670 × 10^−1^
F16	9.1510 × 10^−1^	1.2614 × 10^0^	6.7380 × 10^−1^	9.9940 × 10^−1^	7.5030 × 10^−1^	1.4111 × 10^1^	5.6660 × 10^−1^	6.8380 × 10^−1^	4.9330 × 10^−1^
F17	7.1150 × 10^−1^	1.2121 × 10^0^	6.0240 × 10^−1^	9.1370 × 10^−1^	1.0718 × 10^0^	1.4722 × 10^1^	4.7700 × 10^−1^	6.1810 × 10^−1^	4.1870 × 10^−1^
F18	6.8490 × 10^−1^	1.1509 × 10^0^	5.5690 × 10^−1^	8.8080 × 10^−1^	6.1910 × 10^−1^	1.4674 × 10^1^	4.6230 × 10^−1^	5.6180 × 10^−1^	3.8660 × 10^−1^
F19	1.4690 × 10^0^	1.3812 × 10^0^	7.9660 × 10^−1^	1.1445 × 10^0^	8.4850 × 10^−1^	1.4617 × 10^1^	6.9460 × 10^−1^	8.3090 × 10^−1^	6.0620 × 10^−1^
F20	2.2828 × 10^0^	1.4905 × 10^0^	8.9780 × 10^−1^	1.1964 × 10^0^	8.6360 × 10^−1^	1.4959 × 10^1^	7.2180 × 10^−1^	9.7150 × 10^−1^	6.4140 × 10^−1^
F21	2.2604 × 10^0^	1.6074 × 10^0^	1.0257 × 10^0^	1.4272 × 10^0^	1.1057 × 10^0^	1.4296 × 10^1^	9.0980 × 10^−1^	1.0835 × 10^0^	8.2640 × 10^−1^
F22	3.3694 × 10^0^	1.7528 × 10^0^	1.1738 × 10^0^	1.5229 × 10^0^	1.2353 × 10^0^	1.4679 × 10^1^	1.0392 × 10^0^	1.2358 × 10^0^	9.6330 × 10^−1^
F23	3.9384 × 10^0^	1.9808 × 10^0^	1.3808 × 10^0^	1.6951 × 10^0^	1.4924 × 10^0^	1.4746 × 10^1^	1.2949 × 10^0^	1.4418 × 10^0^	1.1654 × 10^0^

**Table 13 biomimetics-08-00441-t013:** Experiment results of RLHGS and other eight superior algorithms on CEC2020.

	F1			F2			F3		
	Avg	Std	Rank	Avg	Std	Rank	Avg	Std	Rank
RLHGS	2.8842 × 10^4^	3.1972 × 10^4^	2	8.4538 × 10^3^	6.5144 × 10^2^	1	1.0688 × 10^3^	3.3354 × 10^1^	1
CS	1.0000 × 10^10^	0.0000 × 10^0^	7	2.0238 × 10^4^	4.8058 × 10^2^	7	2.5278 × 10^3^	1.8830 × 10^2^	5
MFO	1.5398 × 10^11^	5.0361 × 10^10^	9	1.7701 × 10^4^	2.1140 × 10^3^	5	5.2734 × 10^3^	1.2197 × 10^3^	9
HHO	4.2402 × 10^8^	5.1004 × 10^7^	5	1.9801 × 10^4^	1.6613 × 10^3^	6	4.2440 × 10^3^	2.2379 × 10^2^	8
SSA	3.3528 × 10^4^	3.0298 × 10^4^	3	1.6415 × 10^4^	1.7661 × 10^3^	4	1.8667 × 10^3^	1.7702 × 10^2^	3
JADE	3.5454 × 10^3^	6.0412 × 10^3^	1	1.2316 × 10^4^	6.0211 × 10^2^	2	1.3513 × 10^3^	1.0439 × 10^2^	2
ALCPSO	7.0851 × 10^5^	2.5719 × 10^6^	4	1.5905 × 10^4^	1.8514 × 10^3^	3	2.0011 × 10^3^	2.5319 × 10^2^	4
SCGWO	5.2861 × 10^10^	9.9659 × 10^9^	8	2.4465 × 10^4^	2.7615 × 10^3^	9	2.9184 × 10^3^	2.4402 × 10^2^	6
RDWOA	2.5547 × 10^9^	2.4393 × 10^9^	6	2.0553 × 10^4^	2.6338 × 10^3^	8	3.4061 × 10^3^	2.5938 × 10^2^	7
	F4			F5			F6		
	Avg	Std	Rank	Avg	Std	Rank	Avg	Std	Rank
RLHGS	3.1514 × 10^3^	3.3690 × 10^2^	1	1.9036 × 10^6^	7.1206 × 10^5^	2	3.6308 × 10^3^	3.5935 × 10^2^	1
CS	4.9160 × 10^3^	1.8578 × 10^2^	4	5.6825 × 10^6^	1.1166 × 10^6^	4	7.0747 × 10^3^	2.9911 × 10^2^	5
MFO	5.9846 × 10^3^	6.5043 × 10^2^	7	4.8669 × 10^7^	4.5316 × 10^7^	8	9.3841 × 10^3^	1.9076 × 10^3^	8
HHO	6.1115 × 10^3^	6.6172 × 10^2^	8	1.8198 × 10^7^	5.6157 × 10^6^	6	8.7679 × 10^3^	9.6870 × 10^2^	7
SSA	4.9150 × 10^3^	4.9846 × 10^2^	3	1.9370 × 10^6^	6.8365 × 10^5^	3	6.6963 × 10^3^	8.9749 × 10^2^	4
JADE	3.9526 × 10^3^	3.3609 × 10^2^	2	1.0424 × 10^5^	7.2950 × 10^4^	1	4.8052 × 10^3^	3.6056 × 10^2^	2
ALCPSO	5.1273 × 10^3^	5.7647 × 10^2^	5	2.0526 × 10^7^	1.3196 × 10^7^	7	5.9489 × 10^3^	6.3801 × 10^2^	3
SCGWO	5.6881 × 10^3^	8.3109 × 10^2^	6	8.5044 × 10^7^	3.3393 × 10^7^	9	1.0681 × 10^4^	9.6922 × 10^2^	9
RDWOA	6.2967 × 10^3^	8.4208 × 10^2^	9	1.7386 × 10^7^	1.0132 × 10^7^	5	8.6443 × 10^3^	1.0865 × 10^3^	6
	F7			F8			F9		
	Avg	Std	Rank	Avg	Std	Rank	Avg	Std	Rank
RLHGS	1.3633 × 10^6^	7.1889 × 10^5^	2	2.3500 × 10^3^	2.0959 × 10^−12^	4	2.6883 × 10^3^	4.8347 × 10^2^	5
CS	2.8415 × 10^6^	5.9766 × 10^5^	4	2.3500 × 10^3^	7.6080 × 10^−9^	7	3.5166 × 10^3^	1.1805 × 10^3^	7
MFO	2.8668 × 10^7^	3.3622 × 10^7^	9	2.3539 × 10^3^	2.6837 × 10^0^	9	6.2667 × 10^3^	1.7959 × 10^2^	9
HHO	8.5364 × 10^6^	2.7549 × 10^6^	7	2.3500 × 10^3^	1.8501 × 10^−12^	1	2.6000 × 10^3^	0.0000 × 10^0^	1
SSA	1.7033 × 10^6^	7.0658 × 10^5^	3	2.3500 × 10^3^	5.2391 × 10^−10^	6	2.6006 × 10^3^	1.8723 × 10^0^	4
JADE	3.4821 × 10^4^	1.4681 × 10^4^	1	2.3500 × 10^3^	2.5461 × 10^−11^	5	2.7754 × 10^3^	6.7185 × 10^2^	6
ALCPSO	6.4144 × 10^6^	5.2462 × 10^6^	5	2.3500 × 10^3^	9.2761 × 10^−07^	8	5.9265 × 10^3^	7.2027 × 10^2^	8
SCGWO	2.5290 × 10^7^	1.0245 × 10^7^	8	2.3500 × 10^3^	1.8501 × 10^−12^	1	2.6000 × 10^3^	0.0000 × 10^0^	1
RDWOA	6.7465 × 10^6^	3.3014 × 10^6^	6	2.3500 × 10^3^	1.8501 × 10^−12^	1	2.6000 × 10^3^	0.0000 × 10^0^	1
	F10								
	Avg	Std	Rank	+/−/=					
RLHGS	3.0507 × 10^3^	1.6231 × 10^2^	4	~					
CS	3.3320 × 10^3^	4.8669 × 10^1^	6	10/0/0					
MFO	1.1700 × 10^4^	5.5656 × 10^3^	9	10/0/0					
HHO	2.7000 × 10^3^	0.0000 × 10^0^	1	7/1/2					
SSA	3.3086 × 10^3^	7.1568 × 10^1^	5	6/1/3					
JADE	3.3464 × 10^3^	7.4316 × 10^1^	7	7/3/0					
ALCPSO	3.4605 × 10^3^	1.3361 × 10^2^	8	9/0/1					
SCGWO	2.7000 × 10^3^	0.0000 × 10^0^	1	7/1/2					
RDWOA	2.7000 × 10^3^	0.0000 × 10^0^	1	7/1/2					

**Table 14 biomimetics-08-00441-t014:** Friedman test results on CEC2020.

	RLHGS	CS	MFO	HHO	SSA	JADE	ALCPSO	SCGWO	RDWOA
Average rank	2.3	5.6	8.2	5.0	3.8	2.9	5.5	5.8	5.0
Overall rank	1	7	9	4	3	2	6	8	4

**Table 15 biomimetics-08-00441-t015:** Wilcoxon signed-rank results on CEC2020.

	CS	MFO	HHO	SSA	JADE	ALCPSO	SCGWO	RDWOA
F1	1.7344 × 10^−6^	1.7344 × 10^−6^	1.7344 × 10^−6^	5.3044 × 10^−1^	2.5967 × 10^−5^	2.6230 × 10^−1^	1.7344 × 10^−6^	1.7344 × 10^−6^
F2	1.7344 × 10^−6^	1.7344 × 10^−6^	1.7344 × 10^−6^	1.7344 × 10^−6^	1.7344 × 10^−6^	1.7344 × 10^−6^	1.7344 × 10^−6^	1.7344 × 10^−6^
F3	1.7344 × 10^−6^	1.7344 × 10^−6^	1.7344 × 10^−6^	1.7344 × 10^−6^	1.7344 × 10^−6^	1.7344 × 10^−6^	1.7344 × 10^−6^	1.7344 × 10^−6^
F4	1.7344 × 10^−6^	1.7344 × 10^−6^	1.7344 × 10^−6^	1.7344 × 10^−6^	3.8822 × 10^−6^	1.7344 × 10^−6^	1.7344 × 10^−6^	1.7344 × 10^−6^
F5	1.7344 × 10^−6^	1.7344 × 10^−6^	1.7344 × 10^−6^	7.9710 × 10^−1^	1.7344 × 10^−6^	1.7344 × 10^−6^	1.7344 × 10^−6^	1.7344 × 10^−6^
F6	1.7344 × 10^−6^	1.7344 × 10^−6^	1.7344 × 10^−6^	1.7344 × 10^−6^	1.7344 × 10^−6^	1.7344 × 10^−6^	1.7344 × 10^−6^	1.7344 × 10^−6^
F7	3.1817 × 10^−6^	1.7344 × 10^−6^	1.7344 × 10^−6^	1.2044 × 10^−1^	1.7344 × 10^−6^	2.3534 × 10^−6^	1.7344 × 10^−6^	1.7344 × 10^−6^
F8	1.7344 × 10^−6^	1.7344 × 10^−6^	6.2500 × 10^−2^	1.7344 × 10^−6^	2.6114 × 10^−7^	1.1123 × 10^−6^	6.2500 × 10^−2^	6.2500 × 10^−2^
F9	1.9729 × 10^−5^	1.7344 × 10^−6^	1.0000 × 10^0^	3.1123 × 10^−5^	2.6770 × 10^−5^	1.7344 × 10^−6^	1.0000 × 10^0^	1.0000 × 10^0^
F10	1.7344 × 10^−6^	1.7344 × 10^−6^	1.2290 × 10^−5^	1.7344 × 10^−6^	1.7344 × 10^−6^	1.7344 × 10^−6^	1.2290 × 10^−5^	1.2290 × 10^−5^

**Table 16 biomimetics-08-00441-t016:** Comparison results of nine algorithms on tension/compression spring design problem.

MAs	Optimal Values of Parameters			Optimum Cost
	d	D	N	
RLHGS	0.051749979	0.358185026	11.20345892	**0.0126653**
IHS [100]	0.051154	0.349871	12.076432	0.0126706
MFO [93]	0.053064	0.390718	9.542437	0.012699
PSO [25]	0.015728	0.357644	11.244543	0.0126747
WOA [101]	0.050451	0.327675	13.219341	0.012694
GSA [33]	0.050276	0.345215	13.52541	0.0126763
INFO [36]	0.051555	0.353499	11.48034	0.012666
SMA [28]	0.05847	0.523420486	6.95166221	0.0160198
SMFO [102]	0.06573	0.32869515	2.629561202	0.0138029

**Table 17 biomimetics-08-00441-t017:** Comparison results of ten algorithms on welded beam design problem.

MAs	Optimal Values of Parameters				Optimum Cost
	h	l	t	b	
RLHGS	0.2015	3.3345	9.03662391	0.20572964	**1.699986**
HGS [60]	0.26	5.1025	8.03961	0.26	2.302076
GSA [33]	0.182129	3.856979	10	0.202376	1.879952
CDE [41]	0.203137	3.542998	9.033498	0.206179	1.733462
HS [103]	0.2442	6.2231	8.2915	0.2443	2.3807
GWO [26]	0.205676	3.478377	9.03681	0.205778	1.72624000
BA [104]	2	0.100000	3.174303	2	1.8181
IHS [100]	0.205730	3.470490	9.036620	0.20573	1.7248
RO [105]	0.203687	3.528467	9.004233	0.207241	1.735344
SIMPLEX [106]	0.2792	5.6256	7.7512	0.2796	2.5307

**Table 18 biomimetics-08-00441-t018:** Comparison results of ten algorithms on pressure vessel design problem.

MAs	Optimal Values of Parameters				Optimum Cost
	$T_{s}$	$T_{h}$	R	L	
RLHGS	0.8125	0.4375	42.0984456	176.6365958	**6059.714335**
ES [107]	0.8125	0.4375	42.098087	176.640518	6059.7456
PSO [25]	0.8125	0.4375	42.091266	176.7465	6061.0777
GA [22]	0.9375	0.5	48.329	112.679	6410.3811
G-QPSO [108]	0.8125	0.4375	42.0984	176.6372	6059.7208
SMA [28]	0.75	50.3125	41.17	193.001	6772.7333
Branch-and-bound [109]	1.125	0.625	47.7	117.71	8129.1036
IHS [100]	1.125	0.625	58.29015	43.69268	7197.73
GA3 [81]	0.812500	0.437500	42.0974	176.6540	6059.9463
CPSO [110]	0.812500	0.437500	42.091266	176.746500	6061.0777

**Table 19 biomimetics-08-00441-t019:** Comparison results of seven algorithms on three-bar truss design problem.

MAs	Optimal Values of Parameters		Optimum Cost
	$x_{1}$	$x_{2}$	
RLHGS	0.788673486	0.408252954	**263.89584338**
CS [92]	0.78867	0.40902	263.9716
MFO [93]	0.788244771	0.409466958	263.8959797
BWOA [98]	0.788666327	0.408273202	263.8958435
GOA [111]	0.788897556	0.40761957	263.8958815
MBA [112]	0.7885650	0.4085597	263.8958522
MVO [34]	0.78860276	0.408453070	263.8958499

## Data Availability

The relevant experimental data are within this paper.

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
