# Peer review of "An Enhanced Hunger Games Search Optimization with Application to Constrained Engineering Optimization Problems"

_biomimetics, 2023, doi:10.3390/biomimetics8050441_

Round 1

Reviewer 1 Report

1.      Authors should predefine all acronyms in the abstract and other sections “e.g. The abbreviation of this variant is RLHGS, which integrates adapted Logarithmic Spiral (OBL-LS)”.

2.      The authors should correct Eq. (7), it should be with exponential function.

3.      How do you select the value of LH in the algorithm?

4.      Flowchart in Figure 1 is not well-organized, where I can find i<N is repeated many times, which confuses the reader.

5.      What is the difference between Xb and Xbest?

1.      The authors should compare the execution time of the proposed algorithm with other algorithms. 

Author Response

Reviewer #1:

  1. Authors should predefine all acronyms in the abstract and other sections “e.g. The abbreviation of this variant is RLHGS, which integrates adapted Logarithmic Spiral (OBL-LS)”.

[Response]:

  1. The authors should correct Eq. (7), it should be with exponential function.

[Response]:

Thank you for your reminder and we apologize for the inadequacy of the description of Eq(7). We have revised the formulas in the new manuscript and thoroughly reviewed all remaining equations. We greatly appreciate your comments and the chance to address errors. Thank you once again for your valuable input!

  1. How do you select the value of LH in the algorithm?

[Response]:

Thank you for your comment. The original paper did not specify the specific value of LH. According to the settings in references " Advanced orthogonal learning and Gaussian barebone hunger games for engineering design " and " An artificial bee bare-bone hunger games search for global optimization and high-dimensional feature selection ", the value of LH in this algorithm is set to 100, which has been pointed out in the text and listed in the Table 8.

  1. Flowchart in Figure 1 is not well-organized, where I can find i<N is repeated many times, which confuses the reader.

[Response]:

Thank you for your comment. “i<N” stands for the termination conditions in different stages. In the new manuscript, to make the algorithm process clearer we have redesigned the flowchart in Figure 1.

  1. What is the difference between and ?

[Response]:

Thank you for your comment. In this paper,  and  have the same meaning, both representing the best position in the current iteration. The error has been modified in the new manuscript. Thank you for pointing out the error carefully.

  1. The authors should compare the execution time of the proposed algorithm with other algorithms.

[Response]:

Thank you for your valuable suggestion. The comparison between our research algorithm and other algorithms in terms of execution time on 23 benchmark functions has been reflected in Table 12. Thank you again for your suggestion.

Reviewer 2 Report

The engineering optimization problems have been selected as a case study in this study through the hunger games search (HGS) algorithm. The verification of experimental working has been conducted through some different studies with the collected results. This study is interesting however there are some drawbacks that the authors should address them to improve this study.

  1. There are some grammatical mistakes which have been found in this study,
  2. In the introduction, the authors have provided the contribution of this study. However, the novelty points of this study must be provided and analyzed since there are a lot of different optimization methods like PSO, GWO, etc. Why the hunger game search (HGS) in this study is selected and studied to deal the problems in engineering.
  3. In figure 1, there are some confused points from inter-connected functional blocks, especially the conditional block: i < N? at there positions, actually it is a functional block. Additionally, the update weight W2(i,j), what is the final results from this updates W2 and W1,… Kindly improve this framework of this study. There are a lot of confused spaces.
  4. In figures 3, 5, 7 they are blurred to observe these collected and its comparison. Kindly improve and provided the clear figures to re-evaluate these collected results.
  5. The significance of this optimization algorithm should be implemented in the meta-heuristics optimization algorithms in the conclusion part.

The grammatical mistakes in English must be corrected by the language editors.

Author Response

Reviewer #2:

  1. There are some grammatical mistakes which have been found in this study.

[Response]:

Thank you for your comment. We have thoroughly reviewed the entire text for grammar issues and made the necessary modifications to ensure its correctness.

  1. In the introduction, the authors have provided the contribution of this study. However, the novelty points of this study must be provided and analyzed since there are a lot of different optimization methods like PSO, GWO, etc. Why the hunger game search (HGS) in this study is selected and studied to deal the problems in engineering.

[Response]:

Thanks for your comment. We mainly consider two aspects when doing engineering optimization.

Firstly, the original HGS algorithm has already been successfully applied to three engineering optimization problems. But its performance is not the best(such as the experiment in section 5.3.2), and there are still many other engineering problems worth studying and discussing. Thus, we planned to solve four engineering problems.

The second is that the mechanism we used has excellent performance in solving local optimization problems. Therefore, we have tried RLHGS on different problems, such as feature selection and image segmentation, and engineering optimization. However, we found that RLHGS has significant advantages in engineering optimization problems. Therefore, we have decided to apply it to engineering optimization.

Meanwhile, in the future, we will apply RLHGS to a wider range of problems, such as energy optimization problems and workshop scheduling problems, etc.

  1. In figure 1, there are some confused points from inter-connected functional blocks, especially the conditional block: i < N? at there positions, actually it is a functional block. Additionally, the update weight W2(i,j), what is the final results from this updates W2 and W1, Kindly improve this framework of this study. There are a lot of confused spaces..

[Response]:

Thank you for your comment. Considering your question, to make the algorithm process clearer, we have redesigned the flowchart in Figure 1 in the new manuscript.

  1. In figures 3, 5, 7 they are blurred to observe these collected and its comparison. Kindly improve and provided the clear figures to re-evaluate these collected results.

[Response]:

Thank you for your suggestion. We have made improvements to Figures 3, 5, 7.

  1. The significance of this optimization algorithm should be implemented in the meta-heuristics optimization algorithms in the conclusion part.

Reviewer 3 Report

Please check English grammar

Author Response

Reviewer #3:

  1. In abstract, the authors said HGS shows good performances in global optimization problems. However, HGS emerges some defects, such as lack of the diversity of individuals, premature convergence, and easy to be stuck at local optimal. To solves them, the authors integrates the adapted Logarithmic Spiral (OBL-LS) strategy and the adapted Rosenbrock Method (RM) into the standard HGS. I want to know why you select Logarithmic Spiral (OBL-LS) strategy and Rosenbrock Method (RM) ? What standard is used?

[Response]:

Thanks for your comments.

Because of the defects of the original HGS, we hope to carry out some innovative work in the exploration and exploitation phase of the algorithm. The adapted Logarithmic Spiral (LS-OBL) is an outstanding strategy to reduce the scope of exploration and mitigate the waste of resources due to the large random scope. The Logarithmic Spiral is an excellent local search technique that does not fail to bounce out of the loop due to invalid searches on multimodal problems and can effectively improve the local search performance of the algorithm.

These are the reasons why we incorporated these two strategies into HGS. And I have added these contents to the abstract and section 1, enabling readers to quickly find the central idea of this study.

Thanks again for your valuable question.

  1. About the proposed algorithm RLHGS, the authors should prove that it is not easily stuck at local optimal.

[Response]:

Thanks for your comments.

First of all, in our study, we have conducted extensive experiments and comparisons to evaluate the performance of RLHGS in terms of avoiding local optima. We compared RLHGS with several other algorithms on various optimization problems, including unimodal, multimodal, and fixed-dimensional functions. And the comparison results, as presented in Tables 9 and 13, demonstrate that RLHGS consistently achieves excellent results across different function types, outperforming other algorithms in terms of optimal values. This indicates that RLHGS possesses robustness and versatility when dealing with optimization problems and is not easily stuck at local optima.

Moreover, the convergence curves depicted in Figures 5 (especially on F12, F13) and 7 further support the claim that RLHGS effectively escapes local optima. These curves show that RLHGS exhibits faster convergence rates and achieves closer-to-optimal solutions compared to other algorithms, especially in the late search stage. In other words, The ability to escape local optima is crucial in ensuring the algorithm's effectiveness in finding global or near-optimal solutions.

Overall, through comprehensive experimental evaluations and comparisons, we have demonstrated that RLHGS has a strong capability to avoid local optima and perform well across a wide range of optimization problems.

Thanks again.

  1. For the optimization algorithm, the efficiency is critical. The authors should compare the time and space complexity of the proposal with other optimization algorithms.

[Response]:

Thank you for your comment. The comparison between RLHGS and other algorithms in terms of execution time has been reflected in Table 12. Thank you again for your suggestion.

  1. In Table 2 of Section 4, the authors used 23 classic benchmark functions try to validate their proposal. Please let us know how they decided the domain for each function. What is the initial value of each benchmark function? What is the unit change value of each benchmark function? Authors should write them clearly.

[Response]:

Thank you for your comment.

23 classical benchmarks test suit consist of unimodal functions, multimodal functions, and fixed-dimensional multi-modal functions. Specifically,  represent high-dimensional problems, including unimodal functions (), a step function with one minimum value (), a nosiy quartic function (), and multimodal functions with multiple local optima ().  are low-dimensional functions with only a few local minima, which enables the assessment of the algorithm's effectiveness in searching for near-global optima. Regarding the initial values and unit change values of these benchmark functions, they are not explicitly defined as they can vary depending on the optimization algorithm being used or the specific implementation requirements.

I have added relevant explanations to section 4.1, detailed information about these benchmark functions can be found in Table 2.

Thanks again for your question!

  1. The authors said According to the data in Table 5 and Table 8, it is obviously that RLHGS got the maximum number of optimal values due to its smaller average value (Avg) and standard deviation (Std) in the whole experiment. However, it is very difficult to understand why RLHGS gives the better performance than other algorithms. Please explain more details

[Response]:

Thanks for your comment. There is indeed a problem with the original description. I would like to provide a more detailed explanation.

First of all, in the proposed algorithm, the adapted Logarithmic Spiral (LS-OBL) and Rosenbrock Method (RM) have significantly improved the performance of HGS. Specifically, LS-OBL effectively alleviates the defects of the classic HGS in exploration by narrowing space properly and increasing the diversity of the solution. RM helps the search agent avoid getting trapped into local optima, ensuring stronger convergence towards global optimal results. The deficiencies in the exploration and exploitation stages of the original algorithm have been alleviated to some extent.

Then, average value (Avg) and standard deviation (Std) are usually used to evaluate the performance of algorithms. Among them, Avg is used to evaluate the global search ability and the quality of the solution. Std is used to evaluate the robustness of the algorithm. The better the algorithm performs in the above aspects, the smaller the values of Avg and Std is.

I have added relevant explanations in section 4.3.

Thank you again for your question.

  1. In Section 6, four real-world constrained benchmark problems are used to validate the proposal. These four problems also belong to a class of classical problems shown in Table 2.Why authors use single section to write? In my opinion, they are not much different essentially. All of them optimize the parameters and use the cost functions to evaluate.

[Response]:

            Thanks for your valuable suggestions. Taking your suggestions into consideration, in the revised manuscript, we have integrated this part into section 5. Thank you again for your comments!

  1. Please check English grammar.

[Response]:

            Thank you for your comment. We have thoroughly reviewed the entire text for grammar issues and made the necessary modifications to ensure its correctness.

Round 2

Reviewer 1 Report

The paper is improved and authors addressed my comments.

The paper is improved and authors addressed my comments.

Author Response

Thanks for accepting our paper.

Reviewer 2 Report

The revised manuscript has been improved after addressing the comments from this reviewer. The current manuscript could be accepted to publish on Biomimetics.

Author Response

Thanks for accepting our paper.

Reviewer 3 Report

Please read the attached file.

minor English problem

Author Response

See attached PDF

Round 3

Reviewer 3 Report

No more comments and suggestions

Minor editing of English language required.